# Lower Bounds for Frank-Wolfe on Strongly Convex Sets

**Jannis Halbey** [1 2] **Daniel Deza** [3] **Max Zimmer** [1 2] **Christophe Roux** [1 2] **Bartolomeo Stellato** [3]
**Sebastian Pokutta** [1 2]

## Abstract

We present a constructive lower bound of $\Omega(1/\sqrt{\varepsilon})$ for Frank–Wolfe (FW) when both the objective and the constraint set are smooth and strongly convex, showing that the known uniform $\mathcal{O}(1/\sqrt{\varepsilon})$ guarantees in this regime are tight. It is known that under additional assumptions on the position of the optimizer, FW can converge linearly. However, it remained unclear whether strong convexity of the set can yield rates *uniformly* faster than $\mathcal{O}(1/\sqrt{\varepsilon})$, i.e., irrespective of the position of the optimizer. To investigate this question, we focus on a simple yet representative problem class: minimizing a strongly convex quadratic over the Euclidean unit ball, with the optimizer on the boundary. We analyze the dynamics of FW for this problem in detail and develop a novel computational approach to construct worst-case FW trajectories, which is of independent interest. Guided by these constructions, we develop an analytical proof establishing the lower bound.

## 1. Introduction

The FW algorithm is a classical first-order method for constrained optimization designed for optimization problems of the form

$$\min_{x \in \mathcal{X}} f(x), \qquad (\mathcal{P})$$

where $\mathcal{X} \subseteq \mathbb{R}^d$ is a compact convex set and $f$ is a convex and smooth function (Frank & Wolfe, 1956; Levitin & Polyak, 1966).

Unlike projected gradient methods, which require projecting the iterates onto $\mathcal{X}$, FW replaces projections with a single

Linear Minimization Oracle (LMO) call per iteration. This property makes FW attractive in large-scale settings where projections are computationally costly. Applications include matrix completion (Garber, 2016), optimal transport (Luise et al., 2019), optimal experiment design (Hendrych et al., 2024), and pruning large language models (Roux et al., 2025), among others.

For general convex and smooth objectives, FW requires at least $\Omega(1/\varepsilon)$ gradient and LMO calls to achieve an $\varepsilon$-optimal solution (Jaggi, 2013; Lan, 2013). However, when the constraint set $\mathcal{X}$ is strongly convex, faster rates are possible.

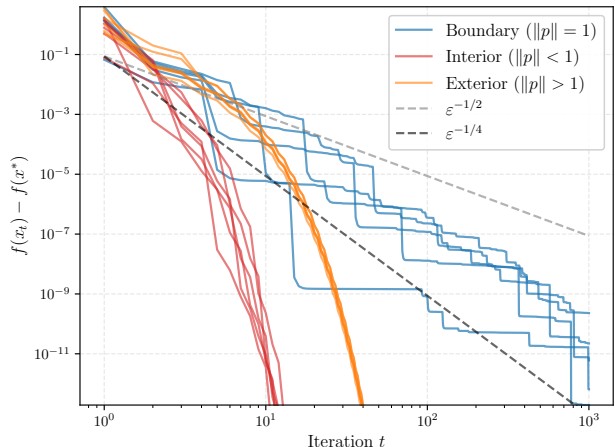

*Figure 1.* Log–log plot of the error versus iteration for FW with exact line search on the function $f(x) = \|x - p\|_2^2$ over the Euclidean unit ball for different positions of the optimizer $p$ with multiple initializations.

In particular, it has long been known that FW converges linearly under additional assumptions on the position of the optimizer, specifically, when the optimizer lies in the interior of $\mathcal{X}$ and $f$ is strongly convex, or when $\|\nabla f(x)\|$ is bounded away from zero on $\mathcal{X}$ (equivalently, the unconstrained optimizer lies outside $\mathcal{X}$) (Wolfe, 1970; Levitin & Polyak, 1966). What remained unclear was whether strong convexity of $\mathcal{X}$ can yield *uniformly* faster rates, without further assumptions on the optimizer location. Garber & Hazan (2015) showed that when both $f$ and $\mathcal{X}$ are strongly convex, FW achieves the uniform rate $\mathcal{O}(1/\sqrt{\varepsilon})$. It is still

[1]AI in Society, Science, and Technology, Zuse Institute Berlin, Berlin, Germany [2]Institute of Mathematics, Technische Universität Berlin, Berlin, Germany [3]Operations Research and Financial Engineering, Princeton University, Princeton, USA. Correspondence to: Jannis Halbey <halbey@zib.de>.

*Proceedings of the 43rd International Conference on Machine Learning*, Seoul, South Korea. PMLR 306, 2026. Copyright 2026 by the author(s).

unclear whether a uniformly faster rate can be achieved in this regime. Figure 1 depicts the empirical convergence on a simple problem. While the instances with optimizer on the boundary converge markedly more slowly than with optimizer in the interior, the observed trajectories appear faster than the worst-case $\mathcal{O}(1/\sqrt{\varepsilon})$ scaling. This motivates the question of whether rates faster than $\mathcal{O}(1/\sqrt{\varepsilon})$ are achievable, or the bound of Garber & Hazan (2015) is tight.

To derive lower bounds, we study the convergence of FW on a simple example: minimizing the quadratic function $f(x) = \|x - p\|_2^2$ over the Euclidean unit ball, where $p$ lies on the boundary of the ball. On this instance, Figure 1 reveals a characteristic behavior: long phases of slow convergence punctuated by intermittent, steep drops in the error. We exploit this structure via a tailored numerical procedure to construct worst-case FW trajectories. Our approach is highly efficient: we can readily compute trajectories with 10,000 iterations or more. Furthermore, it departs significantly from general-purpose worst-case search methods such as the Performance Estimation Problem (PEP) framework: instead of solving a global worst-case program, it leverages problem-specific dynamical structure to synthesize long, high-precision worst-case sequences. We expect this computational approach to be of independent interest for discovering and certifying lower bounds in other settings where long-horizon behavior is decisive.

Based on our numerical results, we show that when both the function and the set are smooth and strongly convex, FW with exact line search or short steps requires at least $\Omega(1/\sqrt{\varepsilon})$ iterations. Notably, the proof of this lower bound follows exactly the same reasoning we use to construct the worst-case sequences. This result resolves a long-standing open question regarding the performance of FW on strongly convex sets, showing that the *uniform* convergence rate of $\mathcal{O}(1/\sqrt{\varepsilon})$ is tight with respect to the precision $\varepsilon$. Moreover, the lower bound is not specific to the particular construction of the Euclidean ball, but can be extended to any ellipsoid. Our result has two interesting implications: First, it establishes that the position of the optimizer genuinely matters for the convergence rate of FW and is not an artifact of prior analyses. Second, since our lower bound applies to smooth sets, it rules out the possibility of a speedup from the smoothness of the set for strongly convex functions.

**Contributions**

1. We provide a detailed analysis of the behavior of FW on our model instance, highlighting the key dynamical phenomena that govern its progress and underpin the observed worst-case trajectories.

2. We develop a numerical procedure based on a forward-backward trajectory construction that computes high-precision worst-case FW sequences over substantially

longer horizons than approaches based on PEP, empirically revealing the $\mathcal{O}(1/\sqrt{\varepsilon})$ scaling. Our numerical procedure might be of independent interest in other settings where high-precision worst-case sequences are needed.

3. We provide an analytical lower bound of $\Omega(1/\sqrt{\varepsilon})$ for FW with exact line search and short steps on smooth and strongly convex sets and functions, showing that the known $\mathcal{O}(1/\sqrt{\varepsilon})$ rates in this setting are tight in their dependence on $\varepsilon$.

**Related Works** Wolfe (1970) showed that FW with exact line search converges linearly if the optimizer lies in the interior of $\mathcal{X}$ and the function $f$ is strongly convex without assuming any conditions on the constraint set other than convexity and compactness. Levitin & Polyak (1966) showed that for strongly convex sets, FW with exact line search or short steps converges linearly if the unconstrained optimizer is bounded away from $\mathcal{X}$. Garber & Hazan (2015) showed that FW with exact line search or short steps converges in $\mathcal{O}(1/\sqrt{\varepsilon})$ if both the function and the constraint set are strongly convex. Wirth et al. (2023) extended this result to open-loop step sizes and Kerdreux et al. (2021b) provided an affine invariant analysis. This result was generalized to uniformly convex sets by Kerdreux et al. (2021a), interpolating between the $\mathcal{O}(1/\sqrt{\varepsilon})$ and $\mathcal{O}(1/\varepsilon)$ rates. There are also other structural assumptions that can be used to improve the convergence rate of FW, for instance polytopal constraints. A wide variety of FW variants, such as the away-step, pairwise, and fully-corrective methods, have been analyzed to achieve improved convergence rates; see Braun et al. (2025) for an overview.

Canon & Cullum (1968) showed an asymptotic lower bound of $\Omega(\varepsilon^{-1/(1+c)})$ for any $c > 0$ for FW with exact line search. Then Jaggi (2013) showed a non-asymptotic lower bound of $\Omega(1/\varepsilon)$ for any feasible step size in the high-dimensional case where $t < d$. Later Lan (2013) extended this lower bound to hold for a family of LMO-based first-order methods. Note, however, that none of these lower bounds apply to the case of strongly convex sets. In fact, except for a lower bound that uses a spectrahedron as constraint set by Jaggi (2013), all of the lower bounds are built on examples with polytopal constraints. The only existing results relevant to our setting are the general lower bounds for first-order methods, which state that the iteration complexity is at least $\Omega(1/\sqrt{\varepsilon})$ and $\Omega(\log(1/\varepsilon))$ for convex functions and strongly convex functions, respectively (Nemirovski & Yudin, 1983). Since the $\mathcal{O}(1/\sqrt{\varepsilon})$ upper bound by Garber & Hazan (2015) requires strong convexity of the function, this left a significant gap between the best known upper and lower bounds for FW on strongly convex sets.

After the original publication of this work, Grimmer & Liu (2026) established a lower bound of $\Omega(1/\sqrt{\varepsilon})$ for LMO-

based methods over strongly convex sets via a zero-chain construction. Their result is complementary to ours: it covers a broader algorithm class, including FW with exact line search and short steps, but is restricted to the high-dimensional regime $T < d$. Moreover, its extension to smooth sets applies only to a reduced algorithm class and only to *modestly* smooth sets, i.e., where the smoothness constant depends on the precision with $L = \Theta(1/\sqrt{\varepsilon})$. In contrast, our result is established based on a smooth and strongly convex instance for any dimension $d \geq 2$, at the cost of being tailored to a narrower algorithmic setting.

The PEP framework enables the systematic construction and search of worst-case instances in a given algorithm and problem class, which can then be used to inspire lower bounds (Taylor et al., 2017; Drori & Teboulle, 2014). Recently, Luner & Grimmer (2024) extended this framework to strongly convex and smooth constraint sets and leveraged these results to build PEPs that incorporate such geometric assumptions. They used this approach to obtain *empirical* lower bound estimates for the performance of FW for strongly convex and smooth sets. However, due to its generality, PEP has a large search space, including the objective function, the (strongly convex and smooth) constraint set geometry, as well as the initialization. The resulting semidefinite programs grow rapidly with the horizon, which typically limits computations to moderate horizons and precisions. In our setting, the numerical estimates reported in Luner & Grimmer (2024) suggest a scaling around $\mathcal{O}(1/\varepsilon^{1/1.44})$ at the horizons they can access. By exploiting the structure of our specific instance, we develop a tailored computational procedure that produces high-precision worst-case sequences over substantially longer horizons, and thereby reveals the $\mathcal{O}(1/\sqrt{\varepsilon})$ scaling, see Figure 7. We find that realizing these worst-case trajectories requires extremely high numerical precision (e.g., constructing a worst-case trajectory over 1000 iterations requires about $10^{-300}$ accuracy), which general-purpose PEP formulations based on large semidefinite programs cannot reliably reach at the horizons needed to reveal the true worst-case rate. Moreover, we turn this high-precision numerical construction into a matching analytical $\Omega(1/\sqrt{\varepsilon})$ lower bound for FW with exact line search and short steps.

**Preliminaries** Throughout this paper, we use $\|\cdot\|$ to denote the Euclidean norm and let $x^* \in \arg\min_{x \in \mathcal{X}} f(x)$ denote an optimizer of ($\mathcal{P}$). We denote by $\partial \mathcal{X}$ the boundary of $\mathcal{X}$ and by $\mathrm{Int}(\mathcal{X})$ the interior of $\mathcal{X}$. A function $f : \mathbb{R}^d \to \mathbb{R}$ is $\mu$-*strongly convex* if

$$f(x) \geq f(y) + \langle \nabla f(y), x - y \rangle + \frac{\mu}{2}\|x - y\|^2$$

---
[1] $\|\nabla f(x)\| \geq c > 0$ for all $x \in \mathcal{X}$
[2] $x^* \in \mathrm{Int}(\mathcal{X})$

*Table 1.* Overview of FW convergence rates. The columns $f$ and $\mathcal{X}$ indicate the assumption of the function and constraint set, respectively, where C means convex and SC to strongly convex. The column $\gamma$ indicates the step-size rule: LS refers to exact line search, ST refers to short steps and OL to open-loop step sizes. The column "rate" indicates the convergence rates.

| | $f$ | $\mathcal{X}$ | $\gamma$ | rate |
|---|---|---|---|---|
| Levitin & Polyak (1966) | C[1] | SC | ST/LS | $\mathcal{O}(\log(\frac{1}{\varepsilon}))$ |
| Wolfe (1970) | SC[2] | C | LS | $\mathcal{O}(\log(\frac{1}{\varepsilon}))$ |
| Garber & Hazan (2015) | SC | SC | ST/LS | $\mathcal{O}(\frac{1}{\sqrt{\varepsilon}})$ |
| Wirth et al. (2023) | SC | SC | OL | $\mathcal{O}(\frac{1}{\sqrt{\varepsilon}})$ |
| e.g. Jaggi (2013) | C | C | OL/LS | $\mathcal{O}(\frac{1}{\varepsilon})$ |
| Nemirovski & Yudin (1983) | SC | C | - | $\Omega(\log(\frac{1}{\varepsilon}))$ |
| Nemirovski & Yudin (1983) | C | C | - | $\Omega(\frac{1}{\sqrt{\varepsilon}})$ |
| **Theorem 6 (Ours)** | SC | SC | ST/LS | $\Omega(\frac{1}{\sqrt{\varepsilon}})$ |

for all $x, y \in \mathbb{R}^d$, and is $L$-*smooth* if

$$f(x) \leq f(y) + \langle \nabla f(y), x - y \rangle + \frac{L}{2}\|x - y\|^2.$$

Let $B_r(x) \stackrel{\mathrm{def}}{=} \{y \in \mathbb{R}^d \mid \|y - x\| \leq r\}$ be the ball of radius $r$ centered at $x$. A closed convex set $\mathcal{X} \subset \mathbb{R}^d$ is $\alpha$-strongly convex if there exists a set $\mathcal{C} \subset \mathcal{X}$ such that $\mathcal{X} = \bigcap_{c \in \mathcal{C}} B_\alpha(c)$. Equivalently, the set $\mathcal{X}$ is strongly convex if, for every point $x \in \partial \mathcal{X}$ and every normal vector $n \in N_{\mathcal{X}}(x) \stackrel{\mathrm{def}}{=} \{n \mid \langle n, y - x \rangle \leq 0 \ \forall y \in \mathcal{X}\}$ with $\|n\| = 1$, one has $\mathcal{X} \subset B_\alpha(x - \alpha n)$ (Goncharov & Ivanov, 2017). Analogously, a closed convex set $\mathcal{X}$ is $\beta$-smooth if, for every $x \in \partial \mathcal{X}$ and every $n \in N_{\mathcal{X}}(x)$ with $\|n\| = 1$, one has $B_\beta(x - \beta n) \subset \mathcal{X}$. All proofs are provided in the appendix in the corresponding section and linked via ($\downarrow$).

## 2. Model Problem Setup

We consider the problem of projecting onto the Euclidean-ball where the projection point $p$ lies on the boundary,

$$\min_{x \text{ s.t. } x \in B_1(0)} \left\{ f(x) \stackrel{\mathrm{def}}{=} \|x - p\|_2^2 \right\}, \quad \|p\| = 1. \quad (\mathcal{P}_B)$$

Both the objective and the constraint set are strongly convex and smooth, with matching parameters $\mu = L = 2$ and $\alpha = \beta = 1$. While this problem is well-conditioned and admits a closed form solution, it turns out to be fundamentally challenging for FW.

We briefly recall the FW update rule:

$$x_0 \in \mathcal{X}, \forall t \in \mathbb{N} \begin{cases} v_t \in \arg\min_{v \in \mathcal{X}} \langle \nabla f(x_t), v \rangle, \\ x_{t+1} = (1 - \gamma_t)x_t + \gamma_t v_t, \end{cases} \quad (1)$$

for some step size $\gamma_t \in [0, 1]$. We begin by showing some basic properties of the FW iterates. First, note that the LMO has a closed-form solution,

$$v_t = -\frac{x_t - p}{\|x_t - p\|} = -\frac{\nabla f(x_t)}{\|\nabla f(x_t)\|} \qquad (2)$$

for $x_t \neq p$. Consequently, the extreme point $v_t$ and thus also the next iterate $x_{t+1}$ lie in the plane spanned by $p$ and $x_t$. By induction, the iterates remain in a two-dimensional invariant subspace defined by $p$ and the initial iterate $x_0$.

**Lemma 1** (Invariant Subspace). [↓] *The iterates $\{x_t\}_{t=0}^{\infty}$ of FW applied to ($\mathcal{P}_B$) satisfy $x_t \in \mathrm{Span}\{p, x_0\}$ for all $t \in \mathbb{N}$.*

Since the iterates remain in the two-dimensional subspace $\mathrm{Span}\{p, x_0\}$, it is convenient to describe them using polar coordinates centered at $p$ within this plane. We define the residual magnitude $r_t$ and the cosine of the angle $\theta_t$ as

$$r_t \overset{\text{def}}{=} \|x_t - p\|, \qquad \theta_t \overset{\text{def}}{=} \frac{\langle x_t - p, p \rangle}{\|x_t - p\|} \in [-1, 0], \qquad (3)$$

where we assume $x_t \neq p$. Generally, if $x_t$ and $p$ are collinear, the problem is one-dimensional and FW can terminate in a single iteration.

The pair $(r_t, \theta_t)$ fully characterizes $x_t$ within this plane.

$$r_{t+1}^2 = \left((1 - \gamma_t) r_t - \gamma_t\right)^2 \\ - 2\gamma_t \left((1 - \gamma_t) r_t - \gamma_t\right) \theta_t + \gamma_t^2, \qquad (4)$$

$$\theta_{t+1} = \frac{\left((1 - \gamma_t) r_t - \gamma_t\right) \theta_t - \gamma_t}{r_{t+1}}, \qquad (5)$$

The recurrences (4) and (5) hold for any step size $\gamma_t \in [0, 1]$. If the step sizes are allowed to be chosen arbitrarily, the algorithm can be made to terminate in at most two iterations from any starting point $x_0 \in B_1(0)$:

**Proposition 2** (Two-step termination). [↓] *There exist step sizes $\gamma_0, \gamma_1 \in [0, 1]$ such that the iterates of FW applied to ($\mathcal{P}_B$) satisfy $x_2 = p$.*

Consequently, any meaningful lower-bound statement for problem ($\mathcal{P}_B$) must assume a specific step-size rule. Classical results by Wolfe (1970) and Levitin & Polyak (1966), which establish linear rates when the unconstrained optimizer lies inside or outside the feasible set, require the step size to be either the *short step* $\gamma_t = \min\left\{1, \frac{\langle \nabla f(x_t), x_t - v_t \rangle}{L \|x_t - v_t\|^2}\right\}$ or the *exact line search* $\gamma_t = \arg\min_{\gamma \in [0,1]} f(x_t + \gamma(v_t - x_t))$.

Thus, to define a FW algorithm achieving the best possible convergence rate for any setting, we fix the step-size rule to be one of these two. Since the objective is an isotropic quadratic function $f(x) = \|x - p\|^2 = r^2$, the exact line

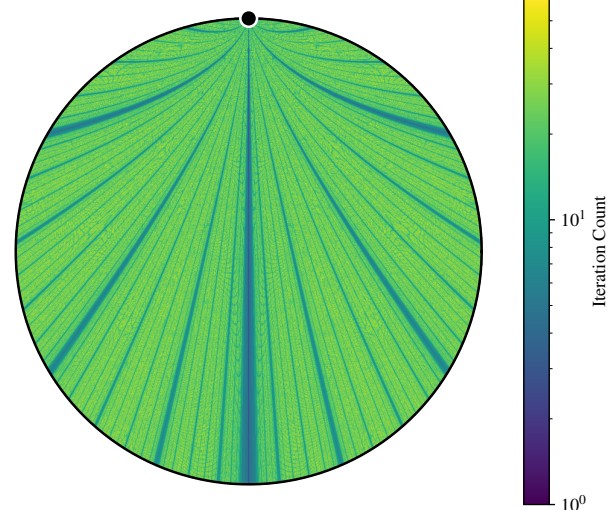

*Figure 2.* Convergence landscape of FW for ($\mathcal{P}_B$) with target at $p = (0, 1)$ denoted by black marker. Colors indicate the number of iterations to reach $10^{-4}$ accuracy from each point in the Euclidean unit ball.

search along the FW direction coincides with the short-step rule. Minimizing $r_{t+1}^2$ as a quadratic in $\gamma$ yields

$$\gamma_t^\star = \frac{r_t(1 + r_t + \theta_t)}{(1 + r_t)^2 + 1 + 2(1 + r_t)\theta_t} \in (0, 1],$$

with equality $\gamma_t^\star = 1$ if and only if $x_t$ and $p$ are collinear, i.e., $\theta_t = -1$.

From this we can directly derive an upper bound on the convergence rate of the residual matching the known rate of $\mathcal{O}(1/\sqrt{\varepsilon})$ from Garber & Hazan (2015).

**Proposition 3.** [↓] *For all $t = 0, \ldots, T$ with $r_t > 0$,*

$$f(x_t) - f(x^*) = r_t^2 \leq \frac{1}{(t + 1/r_0)^2} \qquad (6)$$

*or equivalently $T = \mathcal{O}(1/\sqrt{\varepsilon})$.*

## 3. Finding Worst-Case Trajectories

Despite the worst-case bound established in Proposition 3, it remains unclear whether this rate is optimal for this problem or merely an artifact of the analysis. In particular, it is unknown whether one can establish faster convergence guarantees $\mathcal{O}(1/\varepsilon^a)$ where $a < \frac{1}{2}$.

To gain further insight into the behavior of FW for this specific problem, we turn to an empirical study of the iterates and simulate the FW dynamics for different initializations $(r_0, \theta_0)$. Figure 2 shows that the convergence behavior for this specific problem is very sensitive to the choice of start point. Notably, there is no clear correlation between distance

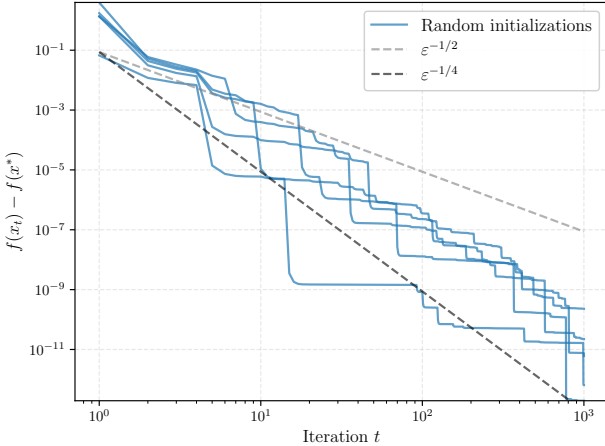

*Figure 3.* Log-log plot of the error versus iteration for Frank–Wolfe with exact line search on the Euclidean unit ball, shown for multiple initializations.

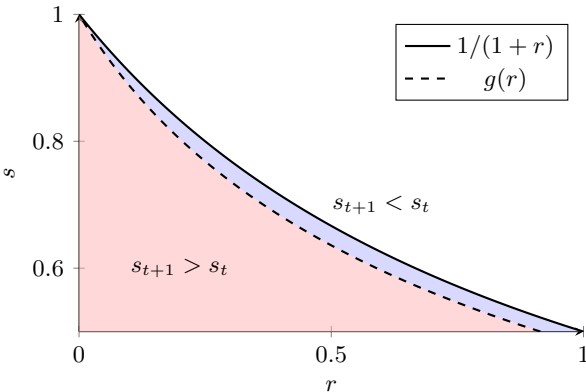

*Figure 4.* Phase diagram of the forward dynamics in the $(r, s)$ state space. The red region indicates the stable phase where the contraction factor increases ($s_{t+1} > s_t$), while the blue region highlights the unstable regime where the contraction improves ($s_{t+1} < s_t$) and jumps occur. The boundary curve $s = g(r)$ separates these two regimes.

to the optimum and convergence speed. In Figure 3, we compare the convergence of the primal gap $f(x_t) - f(x^*)$ for a range of initializations. Across all tested initial conditions, we observe that the empirical rates are closer to $\mathcal{O}(\varepsilon^{-1/4})$ than to $\mathcal{O}(\varepsilon^{-1/2})$, suggesting that the guarantee given in Proposition 3 might not be tight.

All initializations demonstrate a persistent characteristic pattern consisting of extended plateaus corresponding to very slow progress followed by abrupt decreases in the primal gap. The iterates spend long periods in regimes with deteriorating contraction rates before undergoing sudden transitions, or *jumps*, to significantly smaller error levels.

Based on this observed behavior, fast convergence is driven by intermittent sudden jumps. Thus, to understand whether a guaranteed faster rate is possible, we investigate the frequency of these jumps. To that end, we derive an alternative parametrization of the iterates, namely $(r_t, s_t)$, where $s_t$ is the *contraction factor* $s_t \stackrel{\text{def}}{=} r_{t+1}/r_t$.

**Proposition 4.** [↓] *For any iteration t, the iterates $x_t$ of the FW algorithm with exact line search applied to problem* $(\mathcal{P}_B)$ *satisfy*

$$s_{t+1}^2 = \frac{r_{t+2}^2}{r_{t+1}^2} = \frac{1 - (1+r_t)^2 s_t^2}{2 - 2s_t - (2 + r_t)r_t s_t^2}, \qquad (7)$$

$$r_{t+1} = s_t r_t.$$

*where $s_t = r_{t+1}/r_t$ is the contraction factor and $r_t = \|x_t - p\|$ is the residual.*

This reparametrization does not have a one-to-one correspondence with the iterates $x_t$. However, we can show that for each iterate $(r, s) \in M$ there exists a feasible point $x \in B_1(0)$ with the respective residual and contraction factor (see Lemma 9 in Section B.1). The set $M$ is defined

as

$$M \stackrel{\text{def}}{=} \{(r, s) \in \mathbb{R}^2 \mid 0 < r \le 2, 0 \le s \le \bar{s}(r)\}$$

with

$$\bar{s}(r) \stackrel{\text{def}}{=} \begin{cases} \frac{1}{1+r} & \text{for } 0 < r \le 1, \\ \sqrt{\frac{2-r}{4}} & \text{for } 1 < r \le 2. \end{cases}$$

With this alternate parametrization we can formally define jumps as iterations $t$ with contraction factors $s_t < \frac{1}{2}$. We can further characterize these iterations by bounding the preceding contraction factor.

**Proposition 5** (Jump characterization). [↓] *If $s_{t+1} < \frac{1}{2}$, then $s_t > \frac{1}{(1+r_t)^2}$. In particular, if additionally $r_t < \sqrt{2} - 1$, then $s_t > s_{t+1}$.*

This result indicates that jumps are triggered only when the preceding contraction factor $s_t$ is sufficiently large and in particular larger than the subsequent contraction factor $s_{t+1}$ if the residual $r_t$ is sufficiently small. This motivates the search for sequences with monotone increasing contraction rates avoiding jumps. In particular, we investigate whether the first jump of the trajectory can be delayed by constructing initializations that preserve this monotonicity for as long as possible.

We define the *stable phase* of the trajectory as the sequence of iterations where the contraction rates are strictly monotone increasing, i.e., $s_0 < s_1 < \cdots < s_k$. This phase corresponds to the iterates moving along a trajectory where the contraction $s_t$ progressively deteriorates. Intuitively, remaining in this phase as long as possible should lead to the worst convergence rates as jumps do not occur within it.

To visualize these dynamics, Figure 4 depicts the phase space $(r, s)$ partitioned by the critical curve $s = g(r)$, which

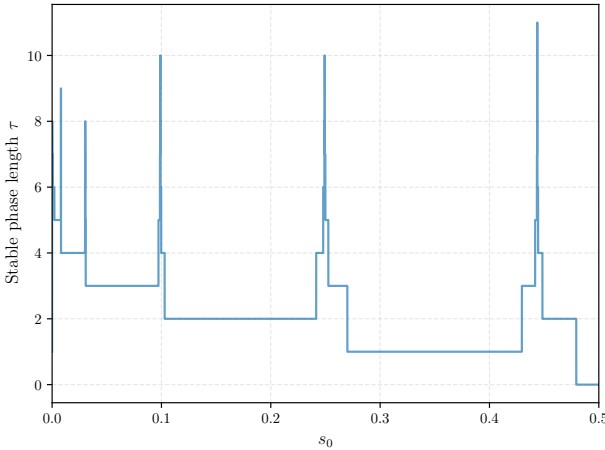

*Figure 5.* Grid search for problem (8), showing the length of the stable phase $\tau$ as a function of $s_0$ for $r_0 = 1$

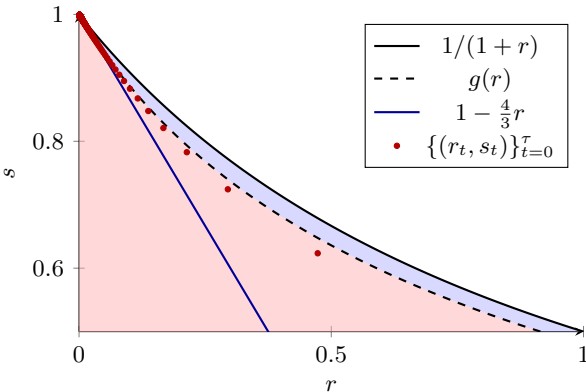

*Figure 6.* Trajectory $\{(r_t, s_t)\}_{t=0}^{\tau}$ of a hard instance in the $(r, s)$ phase space. The affine curve $1 - \frac{4}{3}r$ is shown as a lower bound, while $g(r)$ is an upper bound on the trajectory.

represents the monotonicity threshold where $s_{t+1} = s_t$. This boundary separates the state space into two distinct regimes: the stable domain, highlighted in red, $s_t < g(r_t)$ where the contraction factor strictly increases ($s_{t+1} > s_t$) leading to deteriorating convergence, and the unstable domain, highlighted in blue, $s_t > g(r_t)$ where the rate improves and can yield sudden jumps. The result in Proposition 5 shows that jumps only occur in the unstable regime.

In order to find a sequence that avoids jumps as long as possible, we look for trajectories that remain in the red region for the maximum duration. Formally, we consider the trajectories determined by (7) given an initial contraction factor $s_0 \in [0, \bar{s}(r_0)]$ for some fixed initial residual $r_0$. Among all such trajectories, we seek the one that maximizes the duration $\tau$ of the stable phase, i.e., the number of iterations until the contraction rate first decreases,

$$
\begin{aligned}
\max_{s_0 \in [0, \bar{s}(r_0)]} \quad & \tau \\
\text{s.t. } & r_0 = c, \\
& (r_t, s_t) \text{ determined by (7)}, \\
& s_t \le s_{t+1}, \quad t = 0, \dots, \tau - 1
\end{aligned}
\tag{8}
$$

### 3.1. Forward Search

To solve (8), we perform a grid search over initial iterates $(r_0, s_0) \in M$. We fix $r_0 = 1$ and sample $s_0$ on a uniform grid in $[0, \frac{1}{1+r_0}]$. For each sample, we run the dynamics (7) until the contraction rate decreases, i.e., until $s_{\tau+1} < s_\tau$, recording the stable phase length $\tau$. Figure 5 shows the length of the stable phase $\tau$ as a function of the initial contraction rate $s_0$ for $r_0 = 1$ and a grid of 10,000 samples.

Across different grid resolutions, we consistently observe a structured pattern: the interval $[0, \frac{1}{1+r_0}]$ contains narrow regimes where starting points yield significantly longer sta-

ble phases. The maximal length of these peaks depends on the grid precision. Specifically, finer grids reveal longer phases. A bisection search (Section B.3) confirms this, suggesting one can find starting points with arbitrarily long stable phases and that (8) may be unbounded. We therefore shift our focus to analyzing these stable segments as worst-case constructions for a lower bound.

Figure 6 shows the trajectory $\{(r_t, s_t)\}_{t=0}^{\tau}$ for a starting point corresponding to a peak in the grid search. The trajectory stays closely below the threshold $g(r)$, which acts as an upper bound, and converges to $(0, 1)$. For small $r$, a linear approximation yields $s \approx 1 - \frac{4}{3}r$, and empirically this affine function serves as a lower bound along the trajectory. Notably, this stable trajectory is persistent across all peaks. Different starting points only differ in the number of steps until reaching it. This suggests the stable trajectory is a worst-case instance for FW convergence.

To prove this, we need a lower bound on $s_t$ for all iterations. While $s \ge 1 - \frac{4}{3}r$ is empirically motivated, it lacks theoretical justification since the trajectory's position depends on initial iterates for which we only have numerical values. This motivates a different approach.

### 3.2. Backward Reconstruction Strategy

While the initial condition of the stable trajectory is unknown, setting $r_{t+1} = r_t$ in (7) we see that the trajectory converges to $(0, 1)$. We therefore construct it *backward* from near this terminal point: instead of searching for an initial state that lands on the stable trajectory, we initialize the system at a state $(r_T, s_T)$ for large $T$ on the stable trajectory and iterate the dynamics *backward* in time to recover the precise initial condition $(r_0, s_0)$. Additionally, the results of the bisection search indicate that the stable trajectory satisfies $s \approx 1 - \frac{4}{3}r$ for sufficiently small $r$. Consequently, we initialize the system at a target terminal error $\varepsilon \ll 1$

**Algorithm 1** Backward-Forward Trajectory Construction.

**Input:** Small $\varepsilon > 0$ (terminal error), threshold $r_{\max}$

1: $(r, s) \leftarrow (\varepsilon, 1 - \frac{4}{3}\varepsilon + 2\varepsilon^2)$        $\diamond$ Initialize
2: **while** $r < r_{\max}$ **do**
3:     $X \leftarrow (1 + r)s^2 - r$
4:     $Y \leftarrow \sqrt{(1 - s^2)(1 - (1 + r)^2 s^2)}$
5:     $r_{\text{prev}} \leftarrow \frac{r}{X + Y}$
6:     $s_{\text{prev}} \leftarrow \frac{r}{r_{\text{prev}}} = X + Y$
7:     $(r, s) \leftarrow (r_{\text{prev}}, s_{\text{prev}})$
8: **end while**

**Output:** Start point $(r, s)$

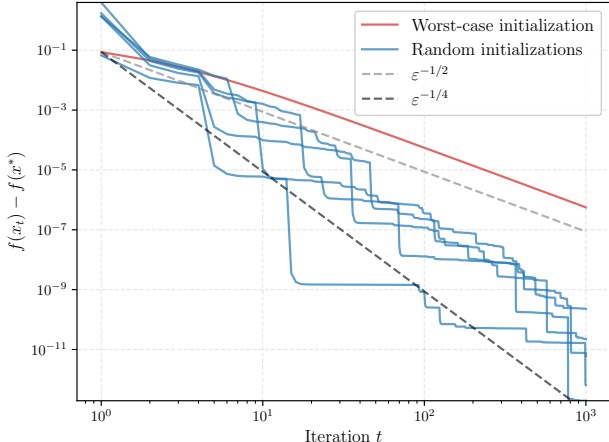

*Figure 7.* Log–log plot of the error versus iteration for Frank–Wolfe with exact line search on the Euclidean unit ball with the worst-case trajectory from Algorithm 1.

representing the state at some large iteration $T$,

$$r_T = \varepsilon, \quad s_T = 1 - \frac{4}{3}r_T + 2r_T^2, \tag{9}$$

where we added a second-order term to $s_T$ for technical reasons (see Section 4 for details). This explicitly places the system on the stable trajectory.

The key part of this approach is the reversibility of the recurrence relations in (7), to which we refer in the following as the *forward dynamics*. The update for $s_{t+1}$ is algebraic, involving a square root in the forward direction. Consequently, recovering the preceding iterate $(r_t, s_t)$ from $(r_{t+1}, s_{t+1})$ involves solving a quadratic equation and thus yields two solutions for $(r_t, s_t)$. However, this does not pose a problem as we just choose the solution that stays on the stable trajectory. The full *backward dynamics* are then given by

$$\begin{aligned}
s_{t-1} &= (1 + r_t)s_t^2 - r_t \\
&\quad + \sqrt{(1 - s_t^2)(1 - (1 + r_t)^2 s_t^2)} \\
r_{t-1} &= \frac{r_t}{s_{t-1}}
\end{aligned} \tag{10}$$

The complete backward reconstruction approach is described in Algorithm 1. Starting with the end point given in (9), we trace the trajectory backward using (10) until the residual $r_t$ is sufficiently large. The constructed trajectory can be then directly verified using the forward dynamics in (7). Beyond this specific application to FW, the backward reconstruction methodology may be of independent interest for establishing lower bounds in other optimization settings where the algorithm's dynamics are (locally) invertible. It sidesteps the need to search over high-dimensional initialization spaces and yields trajectories directly amenable to analytical characterization.

We execute Algorithm 1 to reconstruct the specific initialization $(r_0, s_0)$ that generates the longest stable trajectory. The convergence profile of this worst-case instance is visualized in Figure 7. As evidenced by the alignment with

the reference slope, the empirical convergence rate follows a $\mathcal{O}(1/\sqrt{\varepsilon})$ rate, numerically motivating our lower bound. The corresponding iterates for both the worst-case sequence and random initializations are shown on Figure 8a and Figure 8b respectively. We note that the worst-case sequence follows a damped harmonic oscillation, in contrast to the erratic trajectories of the other initializations, even those close to the worst-case initialization.

We note that the constructed jump-free sequence is not guaranteed to be the global worst-case instance. While we do not preclude the existence of a slower trajectory containing intermittent jumps, this construction proves sufficient to establish the lower bound, as we will show in the next section. Rather than solving a global worst-case program as in PEP, we exploit the problem-specific forward/backward dynamics to explicitly construct a concrete long-horizon worst-case trajectory for our model instance. Note that high-precision arithmetic is instrumental for our approach, as small errors in the backward pass quickly break the stable phase. Therefore, we use the `BigFloat` type with 1000 bits of precision in our Julia implementation. The resulting recursion computes $10^3$ iterations in under 10 seconds on a laptop and readily extends to longer runs. Our code is publicly available on GitHub. [3]

## 4. Lower Bound

In this section, we derive a lower bound for the convergence rate of the FW algorithm applied to $(\mathcal{P}_B)$. Our proof follows exactly the numerical backward construction strategy in Algorithm 1 for finding a start point on the stable trajectory.

---

[3] https://github.com/ZIB-IOL/FW-lower-bound-for-strongly-convex-sets

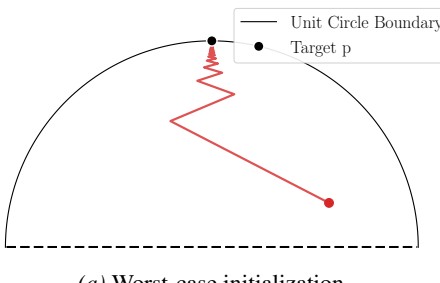

*(a)* Worst-case initialization

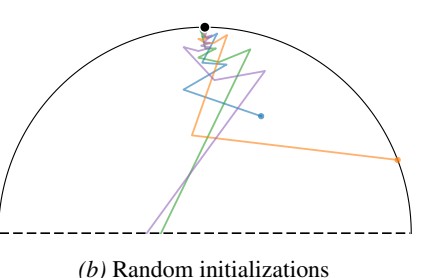

*(b)* Random initializations

*Figure 8.* Iterates of Frank—Wolfe with exact line search on the Euclidean unit ball, showing the worst-case trajectory from Algorithm 1 in Figure 8a and random initializations in Figure 8b, shown on semicircles.

We begin with the end point as defined in (9) and use the backward dynamics to recover the full trajectory. The key idea of our proof is to show that all iterates $(r_t, s_t)$ stay on a stable trajectory strictly above the line $s = 1 - \frac{4}{3}r$ as seen in the numerical experiments. Considering the backward computation of $s_t$ on line 6 of Algorithm 1, one needs to find lower bounds for both $X$ and $Y$ in order to lower bound $s_{\text{prev}}$ and hence also upper and lower bounds on $s$. Consequently, we need both upper and lower bounds for the contraction rates to guarantee that the iterates stay on the stable trajectory after a backward pass.

Our numerical results show that the stable trajectory lies exactly between the $s = 1 - \frac{4}{3}r$ line and the monotonicity threshold $g(r)$. Expanding the latter yields that $g(r) = 1 - \frac{4}{3}r + \mathcal{O}(r^2)$ and thus agrees with the lower bound up to first order. Consequently, any affine linear upper bound larger than $1 - \frac{4}{3}r$ would place the trajectory above $g(r)$ for sufficiently small $r$. Therefore, to make the bounds tight, we add a quadratic term to the approximation of $s_t$,

$$s_t = 1 - \frac{4}{3}r_t + c_t r_t^2. \tag{11}$$

Plugging this in the backward dynamics in Algorithm 1 directly yields bounds for $c_{t-1}$. We show that if $c_t$ lies in the interval $[1, \frac{5}{2}]$, then $(s_{t-1}, r_{t-1})$ satisfies (11) with $c_{t-1} \in [1, \frac{5}{2}]$, see Lemma 15 in Section C. This result demonstrates exactly the preservation of stability of the iterates. In particular, starting from the end point with $c_T = 2$, induction yields that the whole trajectory stays on this

stable trajectory, i.e., satisfies (11) with $c_t \in [1, \frac{5}{2}]$.

As $c_t > 0$, we get directly from (11) that $s \geq 1 - \frac{4}{3}r$ holds for all iterates $(r_t, s_t)$. Rearranging and telescoping then yields

$$r_t \geq \frac{r_0}{1 + \frac{8}{3}r_0 t} \tag{12}$$

for all $t = 0, \ldots, T$.

Finally, we can derive the lower bound for the primal gap directly from the lower bound on the residuals. The result extends to strongly convex functions with arbitrary $\mu > 0$ and sets with arbitrary diameter $D > 0$, by scaling the objective and the constraint set, respectively.

**Theorem 6** (Lower Bound). [↓] *For any $d \geq 2$, $\mu > 0$ and $D > 0$, there exists a compact, smooth, and strongly convex set $\mathcal{X} \subset \mathbb{R}^d$ with $\text{diam}(\mathcal{X}) = D$, a $\mu$-strongly convex and smooth function $f : \mathbb{R}^d \rightarrow \mathbb{R}$, such that for all $T \in \mathbb{N}$ there exists a starting point $x_0 \in \mathcal{X}$ such that the sequence $\{x_t\}_{t=0}^T$ produced by Frank–Wolfe with short steps (or exact line search) satisfies*

$$f(x_t) - f(x^*) \geq \Omega\left(\frac{\mu D^2}{t^2}\right) \tag{13}$$

*for all $t = 1, \ldots, T$ or equivalently, $T = \Omega\left(\sqrt{\mu D^2/\varepsilon}\right)$.*

This result settles the question of whether FW with short steps or exact line search can achieve a convergence rate uniformly faster than $\mathcal{O}(1/\sqrt{\varepsilon})$ for strongly convex sets. While linear convergence for FW is possible in some settings, our lower bound establishes that improving the upper bound uniformly is not possible: the upper bound on the convergence rate by Garber & Hazan (2015) is tight up to a constant factor with respect to the precision $\varepsilon$.

Moreover, due to the affine-invariance of exact line search FW, our lower bound can be directly extended from the problem ($\mathcal{P}_B$) to a class of strongly convex quadratics over ellipsoid sets, see Corollary 17 in Section C.

As shown in Lemma 1, the iterates of FW applied to ($\mathcal{P}_B$) remain in a two-dimensional invariant subspace. Consequently, the lower bound is independent of the problem dimension $d$. Thus, unlike the lower bound in Jaggi (2013) on the simplex or the general lower bounds in Nemirovski & Yudin (1983), this result is not limited to the high-dimensional setting, where $t < d$. Our lower bound is universal for both the high and low-dimensional settings.

Furthermore, note that while the upper bound of Garber & Hazan (2015) requires only that the constraint set is strongly convex, our lower bound holds even when the constraint set is smooth. Therefore, additionally assuming that the set is smooth cannot improve the convergence rate of $\mathcal{O}(1/\sqrt{\varepsilon})$.

On the other hand, numerical experiments show that inaccurate initializations or perturbations of the step-size rule can

cause the iterates to leave the stable trajectory, potentially leading to faster convergence rates. More generally, our lower bound is specific to vanilla FW, hence variants specifically designed for strongly convex sets may circumvent this lower bound.

## 5. Conclusion and Future Work

We studied the worst-case behavior of the FW algorithm on strongly convex and smooth constraint sets through a concrete and geometrically tractable test case: quadratic minimization over the Euclidean unit ball where the optimizer lies on the boundary. Then, we designed a specialized numerical procedure to construct high-precision worst-case sequences over long horizons, leveraging the fact that the iterates remain in an invariant two-dimensional subspace and a detailed characterization of FW's stable slow-convergence trajectories and intermittent jumps. Guided by this construction, we proved an analytical lower bound showing that FW with exact line search or short steps may require $T = \Omega(1/\sqrt{\varepsilon})$ iterations, establishing that the known $\mathcal{O}(1/\sqrt{\varepsilon})$ upper bound is tight w.r.t. the precision $\varepsilon$.

**Future work** Several directions remain open. First, a more fine-grained analysis that quantifies the dependence on problem parameters, such as the smoothness and strong convexity constants of the function and the constraint set. Second, the setting of convex (but not strongly convex) functions still leaves a gap between the upper bound of $\mathcal{O}(1/\varepsilon)$ and the lower bound of $\Omega(1/\sqrt{\varepsilon})$. Third, we observed in numerical experiments that perturbations to the step size and the initialization can cause the iterates to leave the stable trajectory, which calls for a smoothed complexity analysis of FW. Fourth, it would be interesting to apply our computational approach to other algorithms and problem classes where long-horizon worst-case behavior is decisive. Finally, it remains to identify whether FW variants tailored to strongly convex sets can provably circumvent the uniform $\Omega(1/\sqrt{\varepsilon})$ lower bound while retaining the projection-free advantages of FW.

## Acknowledgements

Funded by the Deutsche Forschungsgemeinschaft (DFG, German Research Foundation) under Germany's Excellence Strategy – The Berlin Mathematics Research Center MATH+ (EXC-2046/1, EXC-2046/2, project ID: 390685689). Bartolomeo Stellato is supported by the NSF CAREER Award ECCS-2239771 and the ONR YIP Award N000142512147.

## Impact Statement

This paper presents work whose goal is to advance the field of convex optimization. There are many potential societal consequences of our work, none of which we feel must be specifically highlighted here.

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

# Contents

## A. Problem setup

**Lemma 1** (Invariant Subspace). [↓] *The iterates $\{x_t\}_{t=0}^{\infty}$ of FW applied to $(\mathcal{P}_B)$ satisfy $x_t \in \mathrm{Span}\{p, x_0\}$ for all $t \in \mathbb{N}$.*

*Proof.* (Lemma 1) We proceed by induction on $t$: By definition, $x_0 \in V$, with $V \overset{\text{def}}{=} \mathrm{Span}\{p, x_0\}$, hence the base case holds.

Now assume that $x_t \in V$ for some $t \geq 0$. For $(\mathcal{P}_B)$, we have by Equation (2) that $v_t = -\frac{x_t - p}{\|x_t - p\|}$ if $x_t \neq p$. Hence $v_t$ is a scalar multiple of $x_t - p$, and therefore $v_t \in \mathrm{Span}\{x_t, p\}$. Since $x_t \in V$ by the induction hypothesis and $p \in V$ by definition, it follows that $\mathrm{Span}\{x_t, p\} \subseteq V$, and thus $v_t \in V$. The FW update is given by $x_{t+1} = (1 - \gamma_t)x_t + \gamma_t v_t$, $\gamma_t \in [0, 1]$. As both $x_t$ and $v_t$ belong to the linear subspace $V$, we conclude that $x_{t+1} \in V$.

If $x_t = p$, the algorithm terminates and the result follows trivially. $\qquad\square$

### A.1. Polar coordinates derivation

We briefly recall the definition of the polar coordinates in Equation (3):

$$r_t \overset{\text{def}}{=} \|x_t - p\|, \qquad \theta_t \overset{\text{def}}{=} \frac{\langle x_t - p, p \rangle}{\|x_t - p\|} \in [-1, 0].$$

The update (1) in terms of the polar coordinates is given by

$$x_{t+1} - p = \left((1 - \gamma_t)r_t - \gamma_t\right)\frac{x_t - p}{r_t} - \gamma_t p. \tag{14}$$

Feasibility $\|x_t\| \leq 1$ is equivalent to $\|p + (x_t - p)\|^2 \leq 1$, which gives $1 + r_t^2 + 2r_t\theta_t \leq 1$ and thus

$$\theta_t \leq -\tfrac{1}{2}r_t. \tag{15}$$

Taking the norm and the inner products with $p$ of (14) gives the compact polar FW recurrence in (4) and (5):

$$r_{t+1}^2 = \left((1 - \gamma_t)r_t - \gamma_t\right)^2 - 2\gamma_t\left((1 - \gamma_t)r_t - \gamma_t\right)\theta_t + \gamma_t^2,$$
$$\theta_{t+1} = \frac{\left((1 - \gamma_t)r_t - \gamma_t\right)\theta_t - \gamma_t}{r_{t+1}}.$$

**Proposition 2** (Two-step termination). [↓] *There exist step sizes $\gamma_0, \gamma_1 \in [0, 1]$ such that the iterates of FW applied to $(\mathcal{P}_B)$ satisfy $x_2 = p$.*

*Proof.* (Proposition 2)

For $\gamma_0 = \frac{r_0}{r_0 + 1}$, the next iterate $x_1$ is collinear with $p$:

$$x_1 = x_0 + \gamma_0(v_0 - x_0) = x_0 + \frac{r_0}{r_0 + 1}\left(-\frac{x_0 - p}{r_0} - x_0\right) = \frac{p}{r_0 + 1}.$$

In particular, we have

$$\theta_1 = \frac{\langle x_1 - p, p \rangle}{\|x_1 - p\|} = \frac{\left(\frac{1}{r_0 + 1} - 1\right)\langle p, p \rangle}{\left|\frac{1}{r_0 + 1} - 1\right| \|p\|} = -\|p\| = -1.$$

In the second iteration we use a full step to reach the optimum. Specifically, using $\gamma_1 = 1$ and (4) we get

$$r_2^2 = ((1 - \gamma_1)r_1 - \gamma_1)^2 - 2\gamma_1((1 - \gamma_1)r_1 - \gamma_1)\theta_1 + \gamma_1^2 = 1^2 - 2(-1)(-1) + 1^2 = 0.$$

$\qquad\square$

### A.2. Dynamics under exact line search

We begin by deriving the solution for the exact line search. For that we rewrite (4)

$$
\begin{aligned}
r_{t+1}^2 &= \big((1-\gamma_t)r_t - \gamma_t\big)^2 - 2\gamma_t\big((1-\gamma_t)r_t - \gamma_t\big)\theta_t + \gamma_t^2 \\
&= (r_t - \gamma_t(r_t+1))^2 - 2\gamma_t(r_t - \gamma_t(r_t+1))\theta_t + \gamma_t^2 \\
&= \big((r_t+1)^2 + 2(r_t+1)\theta_t + 1\big)\gamma_t^2 - 2r_t(1+r_t+\theta_t)\gamma_t + r_t^2.
\end{aligned}
\tag{16}
$$

Since $r_{t+1}^2$ is a quadratic in $\gamma_t$, we can obtain the minimizer directly

$$
\gamma_t^* = \frac{r_t(1+r_t+\theta_t)}{(r_t+1)^2 + 2(r_t+1)\theta_t + 1}.
$$

Substituting $\gamma_t^*$ back in (16) yields

$$
\begin{aligned}
r_{t+1}^2 &= \frac{r_t^2(1+r_t+\theta_t)^2}{(r_t+1)^2 + 2(r_t+1)\theta_t + 1} - 2\frac{r_t^2(1+r_t+\theta_t)^2}{(r_t+1)^2 + 2(r_t+1)\theta_t + 1} + r_t^2 \\
&= \frac{-r_t^2(1+r_t+\theta_t)^2 + r_t^2\big((r_t+1)^2 + 2(r_t+1)\theta_t + 1\big)}{(r_t+1)^2 + 2(r_t+1)\theta_t + 1} \\
&= \frac{-r_t^2\big((1+r_t)^2 + 2(1+r_t)\theta_t + \theta_t^2\big) + r_t^2\big((r_t+1)^2 + 2(r_t+1)\theta_t + 1\big)}{(r_t+1)^2 + 2(r_t+1)\theta_t + 1} \\
&= \frac{r_t^2(1-\theta_t^2)}{(r_t+1)^2 + 2(r_t+1)\theta_t + 1}.
\end{aligned}
\tag{17}
$$

Next we consider $r_{t+1}\theta_{t+1}$. Using (5)

$$
\begin{aligned}
r_{t+1}\theta_{t+1} &= \big((1-\gamma_t)r_t - \gamma_t\big)\theta_t - \gamma_t \\
&= r_t\theta_t - \gamma_t(1+\theta_t+r_t\theta_t) \\
&= r_t\theta_t - \frac{r_t(1+r_t+\theta_t)(1+\theta_t+r_t\theta_t)}{(r_t+1)^2 + 2(r_t+1)\theta_t + 1} \\
&= \frac{r_t\theta_t\big((r_t+1)^2 + 2(r_t+1)\theta_t + 1\big) - r_t(1+r_t+\theta_t)(1+\theta_t+r_t\theta_t)}{(r_t+1)^2 + 2(r_t+1)\theta_t + 1} \\
&= \frac{r_t\theta_t\big((r_t+1)^2 + 2(r_t+1)\theta_t\big) - r_t\big((1+r_t) + (1+r_t)^2\theta_t + \theta_t^2(1+r_t)\big)}{(r_t+1)^2 + 2(r_t+1)\theta_t + 1} \\
&= \frac{r_t(1+r_t)(\theta_t(r_t+1) + 2\theta_t^2 - 1 - \theta_t(r_t+1) - \theta_t^2)}{(r_t+1)^2 + 2(r_t+1)\theta_t + 1} \\
&= \frac{r_t(1+r_t)(\theta_t^2 - 1)}{(r_t+1)^2 + 2(r_t+1)\theta_t + 1} \\
&= -\frac{r_t+1}{r_t}r_{t+1}^2
\end{aligned}
$$

Dividing by $r_{t+1} > 0$ yields

$$
\theta_{t+1} = -\frac{r_t+1}{r_t}r_{t+1}.
\tag{18}
$$

**Proposition 3.** [↓] *For all $t = 0, \ldots, T$ with $r_t > 0$,*

$$
f(x_t) - f(x^*) = r_t^2 \le \frac{1}{(t+1/r_0)^2}
\tag{6}
$$

*or equivalently $T = \mathcal{O}(1/\sqrt{\varepsilon})$.*

*Proof.* (Proposition 3) Taking the absolute values of (18), yields $1 \ge |\theta_{t+1}| = \frac{1+r_t}{r_t}r_{t+1}$, which is equivalent to

$$
r_{t+1} \le \frac{r_t}{1+r_t}.
\tag{19}
$$

Equality holds if and only if $|\theta_{t+1}| = 1$. Equation (19) is equivalent to $\frac{1}{r_{t+1}} \ge \frac{1}{r_t} + 1$ and telescoping yields (6). $\square$

## B. Finding Worst-Case Trajectories

### B.1. Forward dynamics

We begin by showing that the dynamics of the polar coordinates can be expressed as a recursion for the residuals $r_t$ and the contraction factor $s_t = \frac{r_{t+1}}{r_t}$ instead of the angle $\theta_t$.

**Proposition 4.** [↓] *For any iteration $t$, the iterates $x_t$ of the FW algorithm with exact line search applied to problem $(\mathcal{P}_B)$ satisfy*

$$s_{t+1}^2 = \frac{r_{t+2}^2}{r_{t+1}^2} = \frac{1 - (1 + r_t)^2 s_t^2}{2 - 2s_t - (2 + r_t)r_t s_t^2}, \tag{7}$$

$$r_{t+1} = s_t r_t.$$

*where $s_t = r_{t+1}/r_t$ is the contraction factor and $r_t = \|x_t - p\|$ is the residual.*

*Proof.* (Proposition 4) We consider (17) at time $t + 1$

$$r_{t+2}^2 = \frac{r_{t+1}^2 (1 - \theta_{t+1}^2)}{(r_{t+1} + 1)^2 + 2(r_{t+1} + 1)\theta_{t+1} + 1}.$$

Using (18) yields

$$1 - \theta_{t+1}^2 = 1 - \frac{(1 + r_t)^2}{r_t^2} r_{t+1}^2$$

and

$$(1 + r_{t+1})^2 + 1 + 2(1 + r_{t+1})\theta_{t+1} = 2 - \frac{2}{r_t} r_{t+1} - \frac{2 + r_t}{r_t} r_{t+1}^2.$$

Combining these equations yields a two-step recursion for the residuals $r_t$

$$r_{t+2}^2 = \frac{r_{t+1}^2 (1 - \frac{(1+r_t)^2}{r_t^2} r_{t+1}^2)}{(r_{t+1} + 1)^2 + 2(r_{t+1} + 1)\theta_{t+1} + 1} = \frac{r_{t+1}^2 (1 - \frac{(1+r_t)^2}{r_t^2} r_{t+1}^2)}{2 - \frac{2}{r_t} r_{t+1} - \frac{2+r_t}{r_t} r_{t+1}^2}.$$

Dividing by $r_{t+1}^2$ and replacing $\frac{r_{t+1}}{r_t}$ with the contraction factor $s_t$ yields the result. $\qquad \square$

We call this recurrence relation the *forward dynamics* and give a formal definition below.

**Definition 7.** *The* forward dynamics *are the function $F : M \to M$ defined by*

$$F(r, s) = \left( rs, \sqrt{\frac{1 - (r + 1)^2 s^2}{2 - 2s - (2 + r)rs^2}} \right), \tag{20}$$

*where the domain $M$ is defined as*

$$M \stackrel{def}{=} \left\{ (r, s) \in \mathbb{R}^2 \;\middle|\; 0 < r \le 1, 0 \le s \le \frac{1}{1+r} \text{ or } 1 < r \le 2, 0 \le s \le \sqrt{\frac{2-r}{4}} \right\}. \tag{21}$$

**Lemma 8.** *The forward dynamics $F : M \to M$ are well defined.*

*Proof.* Let $(r, s) \in M$ and write $F(r, s) = (r_+, s_+)$. Considering the definition of $M$, we note that $\sqrt{\frac{2-r}{4}} \le \frac{1}{1+r}$ for $r \in [1, 2]$. Thus we have $0 \le s \le \frac{1}{1+r}$. Hence the numerator in (20) satisfies

$$1 - (1 + r)^2 s^2 \ge 0.$$

For the denominator $d(s) \overset{\text{def}}{=} 2 - 2s - (2 + r)rs^2$ we note that it is strictly decreasing in $s \geq 0$. Therefore, for all $s \in \left[0, \frac{1}{1+r}\right]$,

$$d(s) \geq d\left(\frac{1}{1+r}\right) = 2 - \frac{2}{1+r} - \frac{(2+r)r}{(1+r)^2} = \frac{r^2}{(1+r)^2} > 0.$$

Consequently, the fraction in (20) is non-negative and $s_+ \in \mathbb{R}$ with $s_+ \geq 0$.

Next, we show $(r_+, s_+) \in M$. Since $r_+ = rs$ and $(r, s) \in M$, we have $r_+ > 0$ and

$$r_+ \leq \begin{cases} \frac{r}{1+r} \leq 1, & 0 < r \leq 1, \\ r\sqrt{\frac{2-r}{4}} \leq 1, & 1 < r \leq 2, \end{cases}$$

where the second inequality uses $r^2(2 - r) \leq 4$ for $r \in [1, 2]$. It remains to prove $s_+ \leq \frac{1}{1+r_+}$. Squaring, using $1 + r_+ > 0$, and substituting the definition of $s_+^2$ gives the equivalent inequality

$$(1 + r_+)^2 \left(1 - (1 + r)^2 s^2\right) \leq 2 - 2s - (2 + r)rs^2.$$

Rearranging yields a perfect square:

$$\left(2 - 2s - (2 + r)rs^2\right) - (1 + rs)^2 \left(1 - (1 + r)^2 s^2\right) = \left(1 - (1 + r)s(1 + rs)\right)^2 \geq 0.$$

Thus $s_+^2 \leq \frac{1}{(1+r_+)^2}$, i.e., $s_+ \leq \frac{1}{1+r_+}$, and since $0 < r_+ \leq 1$ we conclude $(r_+, s_+) \in M$. $\qquad\square$

While the function $F$ is defined for a larger domain than $M$, we will show that the iterates $(r_t, s_t) \in M$ correspond to a feasible point $x \in B_1(0)$. For that we need to show that we can reconstruct the angle $\theta$ for a given residual $r$ and contraction rate $s$.

**Lemma 9.** *For any $(r, s) \in M$, there exists a feasible point $x \in B_1(0)$ with $\|x - p\| = r$ and $\frac{\|x' - p\|}{\|x - p\|} = s$, where $x'$ is the next iterate of a FW step starting at $x$.*

*Proof.* Dividing Equation (17) by $r_t^2$ yields the following relation between the contraction rate $s$ and the angle $\theta$

$$s^2 = \frac{1 - \theta^2}{(r + 1)^2 + 2(r + 1)\theta + 1},$$

where we have dropped the indices for simplicity. Rearranging yields a quadratic in $\theta$,

$$\theta^2 + 2s^2(1 + r)\theta + s^2((1 + r)^2 + 1) - 1 = 0,$$

whose discriminant is

$$\Delta = 4s^4(1 + r)^2 - 4(s^2((1 + r)^2 + 1) - 1) = 4(s^2 - 1)\left(s^2(1 + r)^2 - 1\right).$$

Due to the exact line search, the residuals are monotone decreasing which yields $s \leq 1$ for the contraction rate. Therefore, the discriminant is non-negative if and only if $s \leq \frac{1}{1+r}$. Since $\sqrt{\frac{2-r}{4}} \leq \frac{1}{1+r}$ for $r \in [0, 2]$, the discriminant is non-negative for all $(r, s) \in M$. In the following, we show that the solution

$$\theta = -s^2(r + 1) - \sqrt{(s^2 - 1)(s^2(1 + r)^2 - 1)} \tag{22}$$

satisfies the feasibility condition $\theta \leq -\frac{r}{2}$, which is equivalent to

$$\frac{r}{2} - s^2(r + 1) \leq \sqrt{(s^2 - 1)(s^2(1 + r)^2 - 1)}. \tag{23}$$

In the case that $-s^2(r + 1) < -\frac{r}{2}$, we are directly done since the condition $s \leq \frac{1}{1+r}$ guarantees that the square root is real and positive. For the case $-s^2(r + 1) \geq -\frac{r}{2}$, we have $s \leq \frac{1}{2}$ for $r \in [0, 1]$. By definition of the set $M$ we have

$s \le \sqrt{\frac{2-r}{4}} \le \frac{1}{2}$ for $r \in [1, 2]$. Consequently, we have $r \le 2 - 4s^2$. In the following, we show that the left-hand side of (23) attains its maximum and the right-hand side its minimum for $r = 2 - 4s^2$. Note that the left-hand side is monotone increasing and the right-hand side decreasing in $r$. Substituting $r = 2 - 4s^2$ in both sides of (23) yields

$$1 - 2s^2 - s^2(3 - 4s^2) = 1 - 5s^2 + 4s^4$$

and

$$
\begin{aligned}
\sqrt{(s^2 - 1)(s^2(1 + 2 - 4s^2)^2 - 1)} &= \sqrt{16s^8 - 40s^6 + 33s^4 - 10s^2 + 1} \\
&= \sqrt{(1 - 5s^2 + 4s^4)^2} \\
&= 1 - 5s^2 + 4s^4.
\end{aligned}
$$

Therefore, Equation (23) holds and thus the reconstructed $\theta$ in (22) is feasible for any $(r, s) \in M$. Furthermore, writing $\theta$ as an inner product of two vectors and using the Cauchy-Schwarz inequality, we get

$$
\begin{aligned}
\theta &= -s^2(r + 1) - \sqrt{(1 - s^2)\,(1 - s^2(1 + r)^2)} \\
&= -\left\langle \left(\frac{s}{\sqrt{1 - s^2}}\right), \left(\frac{s(r + 1)}{\sqrt{1 - s^2(1 + r)^2}}\right) \right\rangle \\
&\ge -\sqrt{s^2 + (1 - s^2)} \cdot \sqrt{s^2(1 + r)^2 + (1 - s^2(1 + r)^2)} \\
&= -1.
\end{aligned}
$$

Together with $r > 0$, we have that $(r, \theta)$ are valid polar coordinates for any $(r, s) \in M$ corresponding to a feasible point $x \in B_1(0)$. □

## B.2. Monotonicity threshold

Setting $s_{t+1} = s_t$ in (7) yields

$$s_t^2 = s_{t+1}^2 = \frac{1 - (1 + r_t)^2 s_t^2}{2 - 2s_t - (2 + r_t)r_t s_t^2},$$

which rearranges to

$$(r(r + 2)s^3 + (r^2 + 2r + 2)s^2 - s - 1)(s - 1) = 0,$$

where we have dropped the indices for simplicity. Since $s = 1$ is only feasible for $r = 0$, we consider only the first term. This is quadratic in $r$ and has two solutions

$$r_{1,2}(s) = -1 \pm \sqrt{1 - \frac{2s^2 - s - 1}{s^2(s + 1)}}.$$

Since $r_2 < 0$, we consider $r_1$ and restrict $s$ to $(0, 1]$. Let $g(r)$ be the implicit solution for the equation $r = r_1(s)$, we call this the *monotonicity threshold*.

**Proposition 5** (Jump characterization). [↓] *If $s_{t+1} < \frac{1}{2}$, then $s_t > \frac{1}{(1+r_t)^2}$. In particular, if additionally $r_t < \sqrt{2} - 1$, then $s_t > s_{t+1}$.*

*Proof.* (Proposition 5) Using (7) we have

$$\frac{1}{2} > s_{t+1} = \sqrt{\frac{1 - (1 + r_t)^2 s_t^2}{2 - 2s_t - (2 + r_t)r_t s_t^2}}.$$

Squaring both sides, rearranging and $s_t < 1$ yields

$$
\begin{aligned}
0 < s_t^2(3r_t^2 + 6r_t + 4) - 2s_t - 2 &< s_t^2(3r_t^2 + 6r_t + 3) + 1 - 2s_t - 2 \\
&< s_t^2(3r_t^2 + 6r_t + 3) - 3s_t = 3s_t(s_t(1 + r_t)^2 - 1).
\end{aligned}
$$

Since $s_t \ge 0$, we get $s_t > \frac{1}{(1+r_t)^2}$. For $r_t < \sqrt{2} - 1$, we have $s_t > \frac{1}{2} > s_{t+1}$. □

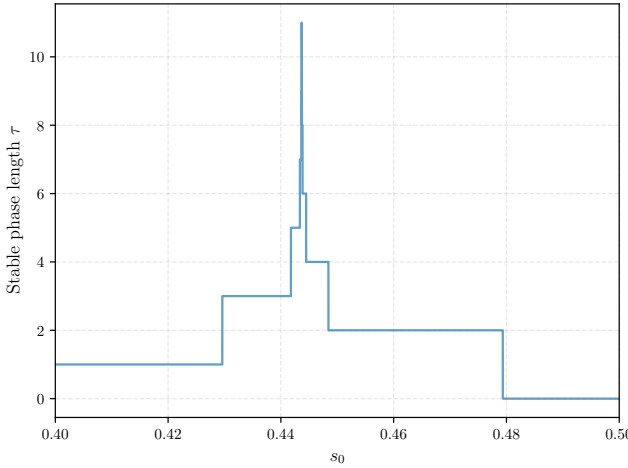

*Figure 9.* Grid search for problem (8), showing the length of the stable phase $\tau$ as a function of $s_0$ for $r_0 = 1$ in the interval $[0.4, 0.5]$

### B.3. Bisection search

In this section, we describe the bisection search, which extends the grid search described in Section 3.1. The purpose of the grid search is to find starting points with long stable phases by directly exploring the forward dynamics in (7). To this end we have fixed the initial residual $r_0 = 1$ and varied the initial contraction $s_0$ on a uniform grid in the admissible interval $[0, \frac{1}{1+r_0}]$.

Recall that the grid search uncovers a remarkably structured pattern. Within the interval $[0, \frac{1}{1+r_0}]$, there are several very narrow yet clearly defined regions where specific starting points lead to much longer stable phases. The maximum length of these peaks is determined by the resolution of the grid search.

To further investigate this pattern, we turn to a more efficient approach without relying on the search over a huge grid. Due to the well-structured pattern and especially the isolated peaks, we can reduce the search space to a single region of long stable phases, for instance $[0.4, 0.5]$ in Figure 9. In these regions, one can empirically observe that the stable phase length $\tau$ increases monotonically in a stair-like fashion to a single maximum before decreasing monotonically again. The stairs on both sides of the maximum have a constant size of 2 (except close to the maximum). Furthermore, we observe that the parity of $\tau$ changes when comparing starting points on the left and right of the maximum. For instance, in the interval $[0.4, 0.5]$, the stable length of starting points smaller than $0.446$ is odd, while it is even for points larger than $0.446$.

This observation motivates a specialized bisection search approach. Starting with an interval $[l, u]$ containing a single peak, we compute the stable phase length $\tau$ for the starting point $m = \frac{l+u}{2}$ in the middle of the interval. We choose the next interval as $[l, m]$ if $\tau(m)$ and $\tau(u)$ have the same parity, otherwise we choose $[m, u]$. This choice guarantees that the starting point with the longest stable phase is consistently contained in the interval. Furthermore, this approach is efficient as it reduces the search space by half in each step.

While these observations remain heuristic, the bisection search turns out to work remarkably well in practice. In high-precision runs, it produces monotone segments that stay on the stable trajectory for arbitrarily long times. In particular, this undermines our initial upper-bound intuition: one cannot uniformly bound the length of the stable phase by controlling the frequency of jumps.

### B.4. Backward dynamics

In this section, we derive the backward dynamics. We start by defining them and then show that they are well defined and that the forward dynamics reverse them.

**Definition 10.** *Let the domain $\widetilde{M}$ be defined as*

$$\widetilde{M} \stackrel{def}{=} \left\{ (r, s) \in \mathbb{R}^2 \mid 0 < r \leq \frac{1}{3},\ 0 \leq s \leq \frac{1}{1+r} \right\}. \tag{24}$$

*The* backward dynamics *are the function* $G : \widetilde{M} \to M$ *defined by*

$$G(r, s) = \left( \frac{r}{X(r,s) + Y(r,s)}, \ X(r,s) + Y(r,s) \right), \tag{25}$$

*where*

$$X(r, s) = (1+r)s^2 - r, \tag{26}$$

$$Y(r, s) = \sqrt{(1 - s^2)(1 - (1+r)^2 s^2)}. \tag{27}$$

*to which we refer as $X$ and $Y$ for simplicity in the following proofs.*

Note that we define the backward dynamics only on the smaller domain $\widetilde{M}$ to ensure that the backward dynamics are well defined. This comes from the technical fact that $X + Y = 0$ for some $(r, s) \in M$. Furthermore, there exist points $x \in B_1(0)$ that cannot be reached by an FW step from any other point $x' \in B_1(0)$. An example of this is the center point $x = 0$.

As a next step, we show that the backward dynamics are well defined and that the forward dynamics reverse them. For that we first prove a lower bound on $X + Y$ for all $(r, s) \in \widetilde{M}$.

**Lemma 11.** *For all $(r, s) \in \widetilde{M}$, we have $Y \in \mathbb{R}$ and $X + Y \geq \frac{5}{12}$.*

*Proof.* First we note that $(r, s) \in \widetilde{M}$ implies $(1 - (1+r)^2 s^2) \geq 1 - \frac{(1+r)^2}{(1+r)^2} = 0$ and $1 - s^2 \geq 1 - \frac{1}{(1+r)^2} \geq 0$. Therefore, $Y$ is well-defined and $Y \geq 0$.

Next, we show that $X + Y \geq \frac{5}{12}$ for all $(r, s) \in \widetilde{M}$. We show that $X + Y$ is monotone decreasing in $s$. For that we compute the derivative of $X + Y$ with respect to $s$:

$$\frac{\partial}{\partial s}(X + Y) = 2(1+r)s + \frac{1}{2\sqrt{(1 - s^2)(1 - (1+r)^2 s^2)}} \cdot \left( -2s(1 - (1+r)^2 s^2) + (1 - s^2)(-2(1+r)^2 s) \right)$$

$$= 2(1+r)s - \frac{s(1 - (1+r)^2 s^2) + (1 - s^2)(1+r)^2 s}{\sqrt{(1 - s^2)(1 - (1+r)^2 s^2)}}$$

$$= 2(1+r)s - \frac{(2 + 2r + r^2)s - 2(1+r)^2 s^3}{\sqrt{(1 - s^2)(1 - (1+r)^2 s^2)}}$$

Therefore, the derivative is negative if and only if

$$2(1+r)s \cdot \sqrt{(1 - s^2)(1 - (1+r)^2 s^2)} \leq (2 + 2r + r^2)s - 2(1+r)^2 s^3.$$

Since both factors in the root are positive for $s \in [0, \frac{1}{1+r}]$, we can use the inequality $\sqrt{ab} \leq \frac{a+b}{2}$ to get:

$$2(1+r)s \cdot \sqrt{(1 - s^2)(1 - (1+r)^2 s^2)} \leq 2(1+r)s \cdot \frac{1 - s^2 + 1 - (1+r)^2 s^2}{2}$$

$$= 2(1+r)s - (1+r)s^3 - (1+r)^3 s^3$$

$$= (2 + 2r)s - (1+r)(2(1+r) + r^2)s^3$$

$$\leq (2 + 2r + r^2)s - 2(1+r)^2 s^3.$$

Consequently, $X + Y$ is monotone decreasing in $s$ and thus attains its minimum at $s = \frac{1}{1+r}$. Together with $r \leq \frac{1}{3}$, we get

$$X + Y = (1+r)s^2 - r + \sqrt{(1 - s^2)(1 - (1+r)^2 s^2)}$$

$$\geq \frac{1+r}{(1+r)^2} - r + \sqrt{\left( 1 - \frac{1}{(1+r)^2} \right)\left( 1 - \frac{(1+r)^2}{(1+r)^2} \right)}$$

$$= \frac{1}{1+r} - r$$

$$\geq \frac{1}{1 + \frac{1}{3}} - \frac{1}{3} = \frac{5}{12}.$$

$\square$

Now we show that the backward dynamics $G$ are well defined and that the forward dynamics $F$ reverse them.

**Lemma 12.** *Let* $F : M \to M, (r, s) \mapsto \left( rs, \sqrt{\frac{1-(r+1)^2 s^2}{2-2s-(2+r)rs^2}} \right)$. *Then the function* $G : \widetilde{M} \to M$ *with*

$$G(r, s) = \left( \frac{r}{X + Y}, X + Y \right)$$

*is well defined and* $F(G(r, s)) = (r, s)$ *holds for all* $(r, s) \in \widetilde{M}$.

*Proof.* Let $(r, s) \in \widetilde{M}$. By Lemma 11, we have $X, Y \in \mathbb{R}$ and $X + Y \geq \frac{5}{12}$, thus $G(r, s)$ is well defined and real. Let $(r', s') = G(r, s)$, we show that $(r', s') \in M$. It is immediate that $s' = X + Y \geq \frac{5}{12} > 0$ and $r' = \frac{r}{X+Y} > 0$ as $r > 0$. Furthermore, we have $r' = \frac{r}{s'} \leq \frac{\frac{1}{3}}{\frac{5}{12}} = \frac{4}{5} < 1$. It is left to show that $s' \leq \frac{1}{1+r'}$ or equivalently $s'(1 + r') \leq 1$. We have already shown in Lemma 11 that $s' = X + Y$ is decreasing in $s$. For $s = 0$ we get $s' \leq -r + 1$ and thus

$$s'(1 + r') = s' \left( 1 + \frac{r}{s'} \right) = s' + r \leq 1 - r + r = 1.$$

In consequence, the function $G : \widetilde{M} \to M$ is well-defined.

Finally, we show that $F$ inverts $G$. Fix $(r, s) \in \widetilde{M}$ and consider a preimage $(r', s')$ under $F$, i.e., assume that $F(r', s') = (r, s)$. By the first coordinate of $F$, we have $r = r's'$ and thus $s' = \frac{r}{r'}$. Consider (20) and substitute $s' = \frac{r}{r'}$

$$s^2 = \frac{1 - (r' + 1)^2 (r/r')^2}{2 - 2(r/r') - (2 + r')r'(r/r')^2}$$

Multiplying both sides by $(r')^2$ and rearranging:

$$s^2 \left( 2(r')^2 - 2rr' - (2 + r')r'r^2 \right) = (r')^2 - (r' + 1)^2 r^2$$

Expanding and collecting powers of $r'$:

$$s^2 \left( 2(r')^2 - 2rr' - 2r'r^2 - (r')^2 r^2 \right) = (r')^2 - r^2((r')^2 + 2r' + 1)$$
$$2s^2(r')^2 - 2rs^2 r' - 2r^2 s^2 r' - r^2 s^2 (r')^2 = (r')^2 - r^2 (r')^2 - 2r^2 r' - r^2$$

Rearranging into standard quadratic form $A(r')^2 + B(r') + C = 0$ with coefficients:

$$A = -r^2 s^2 + r^2 + 2s^2 - 1$$
$$B = -2r^2 s^2 + 2r^2 - 2rs^2$$
$$C = r^2$$

Note that $B = -2rX$ and $A = X^2 - Y^2$,

$$
\begin{aligned}
X^2 - Y^2 &= \left( (1 + r)s^2 - r \right)^2 - (1 - s^2)\left( 1 - (1 + r)^2 s^2 \right) \\
&= ((1 + r)^2 s^4 - 2r(1 + r)s^2 + r^2 - 1 + s^2 + (1 + r)^2 s^2 - (1 + r)^2 s^4 \\
&= s^2 \left( -2r(1 + r) + 1 + (1 + r)^2 \right) - 1 + r^2 \\
&= s^2 \left( 2 - r^2 \right) - 1 + r^2 \\
&= A
\end{aligned}
$$

Therefore, we have

$$(X - Y)(X + Y)r^2 - 2rXr' + r^2 = ((X + Y)r' - r)((X - Y)r' - r) = 0,$$

which yields two roots:

$$r_- = \frac{r}{X+Y} = \frac{r}{(1+r)s^2 - r + \sqrt{(s^2-1)((1+r)^2 s^2 - 1)}} \tag{28}$$

$$r_+ = \frac{r}{X-Y} = \frac{r}{(1+r)s^2 - r - \sqrt{(s^2-1)((1+r)^2 s^2 - 1)}} \tag{29}$$

This shows that the forward dynamics are not injective and there are actually two points $(r_-, \frac{r}{r_-})$ and $(r_+, \frac{r}{r_+})$ that lead to $(r,s)$. Recall that by definition of the backward dynamics,

$$(r', s') = G(r,s) = \left( \frac{r}{X+Y}, X+Y \right) = (r_-, r/r_-),$$

i.e., $G$ selects the branch with denominator $X+Y$. This branch is exactly the one used in the backward reconstruction algorithm and is the branch for which our invariant region is propagated in the subsequent analysis. Therefore, $F(G(r,s)) = F(r', s') = (r,s)$ by construction. $\qquad\square$

**Lemma 13.** *We have $s_t \geq s_{t-1}$ if and only if*

$$(1 + s_t) r_t (2s_t + r_t) \geq (1 - s_t)(2s_t + 1). \tag{30}$$

*Proof.* Using $s_{t-1} = X + Y$, we get directly that $s_t > s_{t-1}$ if and only if $(s_t - X)^2 - Y^2 > 0$. Expanding and grouping by powers of $r$ yields:

$$
\begin{aligned}
(s_t - X)^2 - Y^2 &= (s_t - (1+r_t)s_t^2 - r_t)^2 - (1 - s_t)^2(1 - (1+r_t)^2 s_t^2) \\
&= s_t^2 + (1+r_t)^2 s_t^4 + r_t^2 - 2(1+r_t)s_t^3 - 2s_t r_t + 2r_t(1+r_t)s_t^2 \\
&\quad - (1 - 2s_t + s_t^2) + (1 - 2s_t + s_t^2)(1+r_t)^2 s_t^2 \\
&= (-2s_t^3 + 3s_t^2 - 1) + 2r_t s_t(1 - s_t^2) + r_t^2(1 - s_t^2) \\
&= -(1 - s_t)^2(2s_t + 1) + (1 - s_t^2)r_t(2s_t + r_t) \\
&= (1 - s_t)\left( -(1 - s_t)(2s_t + 1) + (1 + s_t)r_t(2s_t + r_t) \right)
\end{aligned}
$$

Since $0 < s_t < 1$, we get $s_t > s_{t-1} > 0$ if and only if (30) holds. $\qquad\square$

## C. Proof of the lower bound

In this section, we prove the lower bound for the FW algorithm applied to $(\mathcal{P}_B)$. We begin with two technical lemmas that help us to bound the trajectory of the backward pass. In Lemma 14, we find upper and lower bounds for the auxiliary variables $X$ and $Y$ defined in (26) and (27) that match up to the second order.

**Lemma 14.** *For $r \in [0, \frac{1}{10}]$ and $s = 1 - \frac{4}{3}r + cr^2$ with $c \in [1, \frac{5}{2}]$, $X + Y$ is well-defined and satisfies the following bounds:*

$$X + Y \geq 1 - \frac{4}{3}r + \left( \frac{11}{9} - \frac{1}{2}c \right) r^2 - 5r^3,$$

$$X + Y \leq 1 - \frac{4}{3}r + \left( \frac{11}{9} - \frac{1}{2}c \right) r^2 + 2r^3.$$

*Proof.* (Lemma 14) First, since $r \in [0, \frac{1}{10}]$ and $c \in [1, \frac{5}{2}]$, we have $cr \leq \frac{1}{4}$, hence

$$s = 1 - \frac{4}{3}r + cr^2 \leq 1 - \frac{4}{3}r + \frac{1}{4}r < 1 - r \leq \frac{1}{1+r},$$

where the last inequality follows from convexity of $\frac{1}{1+r}$ and $1 - r$ being its tangent at $r = 0$. Therefore, we have $(r,s) \in \widetilde{M}$ and $X + Y$ is well-defined by Lemma 11.

Next we derive the bounds for $X$. For $s = 1 - \frac{4}{3}r + cr^2$, we get

$$X = (1+r)s^2 - r = 1 - \frac{8}{3}r + \left(2c - \frac{8}{9}\right)r^2 + \left(\frac{16}{9} - \frac{2}{3}c\right)r^3 + \left(c^2 - \frac{8}{3}c\right)r^4 + c^2 r^5$$

$$\leq 1 - \frac{8}{3}r + \left(2c - \frac{8}{9}\right)r^2 + \frac{10}{9}r^3 + \frac{25}{4}r^5$$

$$\leq 1 - \frac{8}{3}r + \left(2c - \frac{8}{9}\right)r^2 + 2r^3,$$

where we have used that

$$\frac{16}{9} - \frac{2}{3}c \leq \frac{10}{9}, \quad c^2 - \frac{8}{3}c < 0 \quad \text{and} \quad c^2 \leq \frac{25}{4}$$

for $c \in \left[1, \frac{5}{2}\right]$. Finally we see that $\frac{10}{9}r^3 + \frac{25}{4}r^5 < 2r^3$ for $r \in \left[0, \frac{1}{10}\right]$. For the lower bound, we consider the third and fourth order terms:

$$\left(\frac{16}{9} - \frac{2}{3}c\right)r^3 + \left(c^2 - \frac{8}{3}c\right)r^4 \geq \left(\frac{16}{9} - \frac{2}{3}c + \frac{c^2}{10} - \frac{8}{30}c\right)r^3 = \left(\frac{1}{10}\left(c - \frac{14}{3}\right)^2 - \frac{2}{5}\right)r^3$$

where we have used that the fourth order term is non-positive for $c \in \left[1, \frac{5}{2}\right]$ and thus we can lower bound by setting $r$ to $\frac{1}{10}$. The resulting quadratic in $c$ is decreasing for $c < \frac{14}{3}$ and thus we can lower bound by setting $c$ to $\frac{5}{2}$. Thus, we get

$$\left(\frac{16}{9} - \frac{2}{3}c\right)r^3 + \left(c^2 - \frac{8}{3}c\right)r^4 \geq \left(\frac{1}{10}\left(\frac{5}{2} - \frac{14}{3}\right)^2 - \frac{2}{5}\right)r^3 = \frac{5}{72}r^3 \geq 0.$$

Together with $c^2 r^5 \geq 0$, we get

$$X \geq 1 - \frac{8}{3}r + \left(2c - \frac{8}{9}\right)r^2.$$

Next we consider the bounds for $Y$ by working with $Y^2$ directly. Using $Y^2 = (1-s^2)(1-(1+r)^2 s^2)$ and $s = 1 - \frac{4}{3}r + cr^2$, one gets by direct expansion

$$Y^2 = P(r,c)^2 + r^4 Q(r,c),$$

where

$$P(r,c) = \frac{4}{3}r + \left(\frac{19}{9} - \frac{5}{2}c\right)r^2,$$

and

$$Q(r,c) = \left(-\frac{307}{27} + \frac{65}{9}c - \frac{9}{4}c^2\right) + r\left(-\frac{256}{81} + \frac{536}{27}c - 6c^2\right)$$
$$+ r^2\left(\frac{256}{81} + \frac{64}{27}c - \frac{49}{3}c^2 + 4c^3\right)$$
$$+ r^3\left(-\frac{256}{27}c + \frac{16}{3}c^2 + \frac{8}{3}c^3\right)$$
$$+ r^4\left(\frac{32}{3}c^2 - \frac{20}{3}c^3 + c^4\right) + r^5\left(-\frac{16}{3}c^3 + 2c^4\right) + r^6 c^4.$$

For simplicity, we write only $P$ and $Q$ without the arguments $r$ and $c$ in the following. On $c \in \left[1, \frac{5}{2}\right]$, the coefficient intervals can be coarsely enclosed by integers:

$$Q \in [-8, -5] + [8, 14]r + [-31, -6]r^2 + [-1, 52]r^3 + [1, 7]r^4 + [-11, -3]r^5 + [1, 40]r^6.$$

Since $r^k \in [0, 10^{-k}]$, we can derive the bounds for $Q$ by interval arithmetic:

$$Q \leq -5 + \frac{14}{10} + \frac{52}{10^3} + \frac{7}{10^4} + \frac{40}{10^6} < 0$$

and

$$Q \geq -8 - \frac{31}{10^2} - \frac{1}{10^3} - \frac{11}{10^5} > -8.4.$$

Hence $Y^2 \leq P^2$, and since $Y \geq 0$ and $P \geq 0$, we obtain $Y \leq P$.

For the lower bound, we compute

$$Y^2 - (P - 5r^3)^2 = r^4 \left( Q + 10 \left( \frac{4}{3} + \left( \frac{19}{9} - \frac{5}{2}c \right) r \right) - 25r^2 \right).$$

Now

$$\frac{4}{3} + \left( \frac{19}{9} - \frac{5}{2}c \right) r \geq \frac{4}{3} + \left( \frac{19}{9} - \frac{5}{2} \cdot \frac{5}{2} \right) \cdot \frac{1}{10} = \frac{331}{360} \geq \frac{9}{10}$$

on the same domain, so

$$Q + 10 \left( \frac{4}{3} + \left( \frac{19}{9} - \frac{5}{2}c \right) r \right) - 25r^2 \geq -8.4 + 10 \cdot \frac{9}{10} - 25 \cdot \frac{1}{10^2} > 0.$$

Therefore $Y^2 \geq (P - 5r^3)^2$, and since $P - 5r^3 \geq 0$ for $r \in [0, \frac{1}{10}]$, we get $Y \geq P - 5r^3$. Altogether,

$$Y \leq \frac{4}{3}r + \left( \frac{19}{9} - \frac{5}{2}c \right) r^2 \qquad \text{and} \qquad Y \geq \frac{4}{3}r + \left( \frac{19}{9} - \frac{5}{2}c \right) r^2 - 5r^3,$$

for $r \in [0, \frac{1}{10}]$. Combining the bounds yields the result. $\qquad \square$

As a next step, we show that the factor of the second-order term, when stating $s$ in terms of $r$, can be bounded to a constant interval.

**Lemma 15.** *For $r \in (0, \frac{1}{10}]$, let $s = 1 - \frac{4}{3}r + cr^2$ with $c \in \left[ 1, \frac{5}{2} \right]$. Then the backward step $(r', s') = G(r, s)$ is well-defined. Furthermore, one has $s' = 1 - \frac{4}{3}r' + c'(r')^2$ with $c' \in \left[ 1, \frac{5}{2} \right]$.*

*Proof.* (Lemma 15) We first note that $s \geq 1 - \frac{4}{3}r \geq 1 - \frac{4}{3} \cdot \frac{1}{10} \geq \frac{1}{2}$ and $s \leq 1 - \frac{4}{3}r + \frac{5}{2}r^2 < \frac{1}{1+r}$ for $r \in (0, \frac{1}{10}]$. Therefore, we have that $(r, s) \in \widetilde{M}$ and Lemma 12 implies that $(r', s') = G(r, s)$ is well-defined.

We move now to the bounds for $c'$. We start by rearranging the expression $s' = 1 - \frac{4}{3}r' + c'(r')^2$ and substituting $s' = X + Y$ and $r' = \frac{r}{X+Y}$ yields

$$c' = \frac{s' + \frac{4}{3}r' - 1}{(r')^2} = \frac{(X+Y)^3 - (X+Y)^2}{r^2} + \frac{4(X+Y)}{3r}.$$

Next we consider the expression $Z = (X+Y)^3 - (X+Y)^2$. One can directly integrate both bounds of Lemma 14 to obtain bounds for $Z$. However, by rewriting $Z$ as $(X+Y)^2(X+Y-1)$ one can obtain more precise bounds:

$$
\begin{aligned}
& (X+Y)^2(X+Y-1) \\
& \leq \left( 1 - \frac{4}{3}r + \left( \frac{11}{9} - \frac{1}{2}c \right) r^2 - 5r^3 \right)^2 \left( -\frac{4}{3}r + \left( \frac{11}{9} - \frac{1}{2}c \right) r^2 + 2r^3 \right) \\
& = 50r^9 + \left( -\frac{5}{2}c + \frac{55}{9} \right) r^8 + \left( -2c^2 + \frac{88}{9}c - \frac{1508}{81} \right) r^7 + \left( -\frac{1}{8}c^3 + \frac{11}{12}c^2 - \frac{697}{54}c + \frac{5759}{729} \right) r^6 \\
& \quad + \left( -c^2 + \frac{71}{9}c - \frac{2230}{81} \right) r^5 + \left( \frac{1}{2}c^2 - \frac{46}{9}c + \frac{1418}{81} \right) r^4 + \left( \frac{8}{3}c - \frac{62}{9} \right) r^3 + \left( -\frac{1}{2}c + \frac{43}{9} \right) r^2 - \frac{4}{3}r \\
& \leq -\frac{4}{3}r + \left( -\frac{1}{2}c + \frac{43}{9} \right) r^2 + \left( \frac{8}{3}c - \frac{62}{9} \right) r^3 + 13r^4 + 4r^8 + 50r^9 \\
& \leq -\frac{4}{3}r + \left( -\frac{1}{2}c + \frac{43}{9} \right) r^2 + \left( \frac{8}{3}c - \frac{62}{9} \right) r^3 + 14r^4.
\end{aligned}
$$

Here we used $c \in \left[1, \frac{5}{2}\right]$ to bound the higher-order coefficients uniformly,

$$\frac{1}{2}c^2 - \frac{46}{9}c + \frac{1418}{81} \leq 13, \quad -\frac{5}{2}c + \frac{55}{9} \leq 4,$$

while the coefficients of $r^5, r^6, r^7$ are non-positive on $\left[1, \frac{5}{2}\right]$, and then bounded the higher-order terms for $r \in \left(0, \frac{1}{10}\right]$. The lower bound follows analogously:

$$(X + Y)^2(X + Y - 1)$$

$$\geq \left(1 - \frac{4}{3}r + \left(\frac{11}{9} - \frac{1}{2}c\right)r^2 + 2r^3\right)^2 \left(1 - \frac{4}{3}r + \left(\frac{11}{9} - \frac{1}{2}c\right)r^2 - 5r^3 - 1\right)$$

$$= -20r^9 + \left(8c - \frac{176}{9}\right)r^8 + \left(-\frac{1}{4}c^2 + \frac{11}{9}c + \frac{1607}{81}\right)r^7 + \left(-\frac{1}{8}c^3 + \frac{11}{12}c^2 - \frac{193}{54}c - \frac{10873}{729}\right)r^6$$

$$+ \left(-c^2 + \frac{71}{9}c - \frac{1222}{81}\right)r^5 + \left(\frac{1}{2}c^2 - \frac{46}{9}c + \frac{1418}{81}\right)r^4 + \left(\frac{8}{3}c - \frac{125}{9}\right)r^3 + \left(-\frac{1}{2}c + \frac{43}{9}\right)r^2 - \frac{4}{3}r$$

$$\geq -\frac{4}{3}r + \left(-\frac{1}{2}c + \frac{43}{9}\right)r^2 + \left(\frac{8}{3}c - \frac{125}{9}\right)r^3 + 7r^4 - 9r^5 - 21r^6 + 20r^7 - 12r^8 - 20r^9$$

$$\geq -\frac{4}{3}r + \left(-\frac{1}{2}c + \frac{43}{9}\right)r^2 + \left(\frac{8}{3}c - \frac{125}{9}\right)r^3 + 5r^4.$$

Here we used $c \in \left[1, \frac{5}{2}\right]$ to bound the higher-order coefficients uniformly,

$$\frac{1}{2}c^2 - \frac{46}{9}c + \frac{1418}{81} \geq 7, \quad -c^2 + \frac{71}{9}c - \frac{1222}{81} \geq -9, \quad -\frac{1}{8}c^3 + \frac{11}{12}c^2 - \frac{193}{54}c - \frac{10873}{729} \geq -21,$$

$$-\frac{1}{4}c^2 + \frac{11}{9}c + \frac{1607}{81} \geq 20, \quad 8c - \frac{176}{9} \geq -12,$$

and then bounded the higher-order terms for $r \in \left(0, \frac{1}{10}\right]$.

Consequently we get

$$c' = \frac{(X + Y)^2(X + Y - 1)}{r^2} + \frac{4(X + Y)}{3r}$$

$$\leq \frac{-\frac{4}{3}r + \left(-\frac{1}{2}c + \frac{43}{9}\right)r^2 + \left(\frac{8}{3}c - \frac{62}{9}\right)r^3 + 14r^4}{r^2} + \frac{4\left(1 - \frac{4}{3}r + \left(\frac{11}{9} - \frac{1}{2}c\right)r^2 + 2r^3\right)}{3r}$$

$$= -\frac{4}{3r} + \left(-\frac{1}{2}c + \frac{43}{9}\right) + \left(\frac{8}{3}c - \frac{62}{9}\right)r + 14r^2 + \frac{4}{3r} - \frac{16}{9} + \left(\frac{44}{27} - \frac{2}{3}c\right)r + \frac{8}{3}r^2$$

$$= 3 - \frac{c}{2} + \left(2c - \frac{142}{27}\right)r + \frac{50}{3}r^2 \leq 3 - \frac{1}{2} + \left(2 - \frac{142}{27}\right)r + \frac{50}{3}r^2 \leq \frac{5}{2},$$

where we have used that $-\frac{c}{2} + \left(2c - \frac{142}{27}\right)r$ is monotone decreasing in $c$ for $r \in \left[0, \frac{1}{10}\right]$ and so we could fix $c = 1$. Further, note that $-\frac{88}{27}r + \frac{50}{3}r^2$ is non-positive for $r \in \left[0, \frac{1}{10}\right]$. For the lower bound, we have that

$$c' = \frac{(X + Y)^2(X + Y - 1)}{r^2} + \frac{4(X + Y)}{3r}$$

$$\geq \frac{-\frac{4}{3}r + \left(-\frac{1}{2}c + \frac{43}{9}\right)r^2 + \left(\frac{8}{3}c - \frac{125}{9}\right)r^3 + 5r^4}{r^2} + \frac{4\left(1 - \frac{4}{3}r + \left(\frac{11}{9} - \frac{1}{2}c\right)r^2 - 5r^3\right)}{3r}$$

$$= -\frac{4}{3r} + \left(-\frac{1}{2}c + \frac{43}{9}\right) + \left(\frac{8}{3}c - \frac{125}{9}\right)r + 5r^2 + \frac{4}{3r} - \frac{16}{9} + \left(\frac{44}{27} - \frac{2}{3}c\right)r - \frac{20}{3}r^2$$

$$= 3 - \frac{c}{2} + \left(2c - \frac{331}{27}\right)r - \frac{5}{3}r^2$$

$$\geq 3 - \frac{5}{4} + \left(5 - \frac{331}{27}\right)r - \frac{5}{3}r^2 \geq 3 - \frac{5}{4} - \frac{196}{270} - \frac{5}{300} = \frac{136}{135} > 1,$$

where we have used that $-\frac{c}{2} + \left(2c - \frac{331}{27}\right)r$ is monotone decreasing in $c$ for $r \in \left[0, \frac{1}{10}\right]$ and so we could fix $c = \frac{5}{2}$. Additionally, $\left(5 - \frac{331}{27}\right)r - \frac{5}{3}r^2$ is monotone decreasing for $r > 0$, so we could fix $r = \frac{1}{10}$. $\qquad\square$

Now we can prove the main result. The idea is to use the two-dimensional recurrence for the residuals $r_t$ and contraction factors $s_t$ that correspond to the actual FW trajectory (see Proposition 4), where points $(r_t, s_t) \in M$ correspond to feasible iterates in $B_1(0)$ by Lemma 9. We then employ the backward reconstruction of a starting point for our lower bound. Starting from a terminal point with small residual $\epsilon$, we iteratively apply the backward dynamics $G$ from Definition 10, which are inverted by the forward dynamics $F$ according to Lemma 12. The key technical contribution is showing via induction, using the bounds in Lemmas 14 and 15, that these backward iterates remain in a controlled region where $s = 1 - \frac{4}{3}r + cr^2$ with $c \in [1, \frac{5}{2}]$. This invariant yields a telescoping bound on the residuals, from which the $\Omega(1/t^2)$ lower bound follows.

**Theorem 6** (Lower Bound). [↓] *For any $d \geq 2$, $\mu > 0$ and $D > 0$, there exists a compact, smooth, and strongly convex set $\mathcal{X} \subset \mathbb{R}^d$ with $\mathrm{diam}(\mathcal{X}) = D$, a $\mu$-strongly convex and smooth function $f : \mathbb{R}^d \to \mathbb{R}$, such that for all $T \in \mathbb{N}$ there exists a starting point $x_0 \in \mathcal{X}$ such that the sequence $\{x_t\}_{t=0}^T$ produced by Frank–Wolfe with short steps (or exact line search) satisfies*

$$f(x_t) - f(x^*) \geq \Omega\left(\frac{\mu D^2}{t^2}\right) \tag{13}$$

*for all $t = 1, \ldots, T$ or equivalently, $T = \Omega\left(\sqrt{\mu D^2/\varepsilon}\right)$.*

*Proof.* (Theorem 6)

First we consider the default problem ($\mathcal{P}_B$) with diameter $D = 2$ and strong convexity parameter $\mu = 2$ and later extend the result to the generalized problem.

We begin by considering iterates in the variables $(r_t, s_t)$, which correspond to the residual and contraction factor along Frank-Wolfe (FW) trajectories. Specifically, by Lemma 9, any pair $(r_t, s_t)$ in the admissible region describes a feasible iterate on the Euclidean ball, allowing us to reconstruct the underlying trajectory of the FW algorithm on the $B_1(0)$ ball.

Before we proceed with the proof, we remind the reader of the domain of the backward dynamics $G$, namely $\widetilde{M}$:

$$\widetilde{M} = \left\{ (r, s) \in \mathbb{R}^2 \,\middle|\, 0 < r \leq \frac{1}{3}, 0 \leq s \leq \frac{1}{1+r} \right\}.$$

We start by applying Algorithm 1 with threshold $r_{\max} = \frac{1}{10}$ to construct a starting point $(r_0, s_0)$ from a given terminal point $(\epsilon, 1 - \frac{4}{3}\epsilon + 2\epsilon^2)$ using the backward dynamics $G$ in (10). For a given target horizon $T \in \mathbb{N}$, we choose

$$\epsilon \stackrel{\text{def}}{=} \frac{1}{10 + T^{\frac{8}{3}}}. \tag{31}$$

Let $(u_0, v_0) = (\epsilon, 1 - \frac{4}{3}\epsilon + 2\epsilon^2)$ be the starting point of the backward reconstruction and define the backward iterates $(u_t, v_t) \stackrel{\text{def}}{=} G^t(u_0, v_0)$. Algorithm 1 terminates when the residual is above the threshold, i.e., at the first index $t$ with $u_t \geq r_{\max} = \frac{1}{10}$. We denote by $\widehat{T}$ the last index such that $u_{\widehat{T}} < r_{\max}$ (so $u_{\widehat{T}+1} \geq r_{\max}$), and consider the backward iterates $(u_t, v_t)_{t=0}^{\widehat{T}}$.

We first show that the backward iterates are well-defined. Since $\epsilon \leq \frac{1}{10} \leq \frac{1}{3}$ and $v_0 = 1 - \frac{4}{3}\epsilon + 2\epsilon^2 \leq \frac{1}{1+\epsilon}$ for $\epsilon \in (0, \frac{1}{10}]$, we have $(u_0, v_0) \in \widetilde{M}$. We continue by induction to show that $(u_t, v_t) \in \widetilde{M}$ for all $t = 0, \ldots, \widehat{T}$. By Lemma 12 we have that $(u_{t+1}, v_{t+1}) = G(u_t, v_t)$ is well-defined and belongs to $M$ for $t = 0, \ldots, \widehat{T} - 1$. Since $u_{t+1} \leq r_{\max} = \frac{1}{10}$, it follows by definition of $M$ that $v_{t+1} \leq \frac{1}{1+u_{t+1}}$ and thus $(u_{t+1}, v_{t+1}) \in \widetilde{M}$.

Next, we relate the sequence length $\widehat{T}$ to the target horizon $T$. Again by induction using that $v_0 = 1 - \frac{4}{3}u_0 + 2u_0^2$ and then iteratively applying Lemma 15, we have that

$$v_t = 1 - \frac{4}{3}u_t + c_t u_t^2 \tag{32}$$

with $c_t \in [1, \frac{5}{2}]$ for $t = 0, \ldots, \widehat{T}$ and in particular $v_t \geq 1 - \frac{4}{3}u_t$. By the backward update rule, we have $u_{t+1} = \frac{u_t}{v_{t+1}}$ and thus

$$\frac{u_t}{u_{t+1}} = v_{t+1} \geq 1 - \frac{4}{3}u_{t+1}.$$

Rearranging yields

$$\frac{1}{u_{t+1}(1 - \frac{4}{3}u_{t+1})} \geq \frac{1}{u_t}.$$

For $t + 1 \leq \widehat{T}$, we have $u_{t+1} \leq \frac{1}{10}$ and thus $\frac{4}{3}u_{t+1} \in \left(0, \frac{1}{2}\right)$. Using that $\frac{1}{1-x} \leq 1 + 2x$ for $x \in \left(0, \frac{1}{2}\right)$ yields

$$\frac{1}{u_{t+1}(1 - \frac{4}{3}u_{t+1})} \leq \frac{1}{u_{t+1}}\left(1 + \frac{8}{3}u_{t+1}\right) = \frac{1}{u_{t+1}} + \frac{8}{3}.$$

Combining both inequalities yields,

$$\frac{1}{u_{t+1}} \geq \frac{1}{u_t} - \frac{8}{3}. \tag{33}$$

Telescoping (33) gives $\frac{1}{u_t} \geq \frac{1}{u_0} - \frac{8}{3}t$ for all $t = 0, \ldots, \widehat{T}$. With the choice (31), we have $\frac{1}{u_0} = \frac{1}{\epsilon} = 10 + \frac{8}{3}T$ and thus $u_T \leq \frac{1}{10}$. By definition of $\widehat{T}$, this implies $T \leq \widehat{T}$.

Define now the starting point $(r_0, s_0)$ as the last backward iterate below the threshold, i.e.,

$$(r_0, s_0) \overset{\text{def}}{=} (u_{\widehat{T}}, v_{\widehat{T}}).$$

This links the backward trajectory to a forward FW trajectory. Since $(u_t, v_t) \in \widetilde{M}$ for all $t = 0, \ldots, \widehat{T}$, Lemma 12 implies that

$$(r_t, s_t) = F^t(r_0, s_0) = F^t(G^{\widehat{T}}(u_0, v_0)) = G^{\widehat{T}-t}(u_0, v_0) = (u_{\widehat{T}-t}, v_{\widehat{T}-t}) \tag{34}$$

for all $t = 0, \ldots, \widehat{T}$, and in particular for all $t = 0, \ldots, T$. By Lemma 9, the trajectory $\{(r_t, s_t)\}_{t=0}^{\widehat{T}} \subset M$ corresponds to feasible points $\{x_t\}_{t=0}^{\widehat{T}} \subset B_1(0)$.

Now we turn to the lower bound on the residuals $r_t$. By (33), we have for all $t = 0, \ldots, \widehat{T} - 1$,

$$\frac{1}{u_t} \leq \frac{1}{u_{\widehat{T}}} + \frac{8}{3}(\widehat{T} - t),$$

and thus

$$u_t \geq \frac{u_{\widehat{T}}}{1 + \frac{8}{3}u_{\widehat{T}}(\widehat{T} - t)}.$$

Together with (34) we get

$$r_t = u_{\widehat{T}-t} \geq \frac{u_{\widehat{T}}}{1 + \frac{8}{3}u_{\widehat{T}}t} = \frac{r_0}{1 + \frac{8}{3}r_0 t}$$

for all $t = 0, \ldots, \widehat{T}$, and in particular for all $t = 0, \ldots, T$.

Next, we show that $r_0$ is bounded away from $0$ uniformly and hence is independent of $T$. Since $u_{\widehat{T}} < \frac{1}{10}$, Lemma 15 implies that the next iterate $(u_{\widehat{T}+1}, v_{\widehat{T}+1})$ is well-defined and satisfies

$$v_{\widehat{T}+1} = 1 - \frac{4}{3}u_{\widehat{T}+1} + c_{\widehat{T}+1}u_{\widehat{T}+1}^2$$

for some $c_{\widehat{T}+1} \in \left[1, \frac{5}{2}\right]$. In particular,

$$v_{\widehat{T}+1} \geq 1 - \frac{4}{3}u_{\widehat{T}+1} + u_{\widehat{T}+1}^2 = \left(u_{\widehat{T}+1} - \frac{2}{3}\right)^2 + \frac{5}{9} \geq \frac{5}{9}.$$

Using $u_{\widehat{T}} = v_{\widehat{T}+1}u_{\widehat{T}+1}$ and $u_{\widehat{T}+1} \geq \frac{1}{10}$ by definition of $\widehat{T}$ yields $r_0 = u_{\widehat{T}} \geq \frac{5}{9} \cdot \frac{1}{10} = \frac{1}{18}$.

Next, $f(x_t) - f(x^*) = \|x_t - p\|^2 = r_t^2$ yields the result for the default model problem $(\mathcal{P}_B)$,

$$f(x_t) - f(x^*) \geq \frac{r_0^2}{(1 + \frac{8}{3}r_0 t)^2} = \Omega\left(\frac{1}{t^2}\right).$$

Finally, we consider the generalized model problem

$$\min_{x \in B_R(0)} \tilde{f}(x) \stackrel{\text{def}}{=} \frac{\mu R^2}{2} f\left(\frac{x}{R}\right) = \min_{x \in B_1(0)} \frac{\mu R^2}{2} f(x).$$

whose objective is $\mu$-strongly convex and whose constraint set has diameter $D = 2R$. Therefore, the result for the generalized problem follows from the result for the default model problem ($\mathcal{P}_B$) by scaling the lower bound by $\mu R^2/2$,

$$\tilde{f}(x_t) - \tilde{f}(x^*) \geq \frac{\mu R^2}{2} \frac{r_0^2}{(1 + \frac{8}{3}r_0 t)^2} = \Omega\left(\frac{\mu R^2}{t^2}\right) = \Omega\left(\frac{\mu D^2}{t^2}\right).$$

$\square$

**Corollary 16** (Monotonicity of contraction rates). [↓] *For any $T \in \mathbb{N}$, there exists an $(r_0, s_0) \in M$ such that*

$$s_{t+1} \geq s_t$$

*for all $t = 0, \ldots, T - 1$.*

*Proof.* (Corollary 16) We consider the iterates $(r_t, s_t)$ constructed in Theorem 6. Due to Lemma 13 it suffices to check $(1 + s)r(2s + r) \geq (1 - s)(2s + 1)$ for $s = 1 - \frac{4}{3}r + cr^2$ where $c \in \left[1, \frac{5}{2}\right]$. Comparing

$$(1 + s)r(2s + r) = \left(2 - \frac{4}{3}r + cr^2\right) r \left(2 - \frac{8}{3}r + 2cr^2 + r\right)$$

$$= 4r - 6r^2 + \frac{20}{9}r^3 + 6cr^3 - \frac{13}{3}cr^4 + 2c^2 r^5$$

and

$$(1 - s)(2s + 1) = \left(\frac{4}{3}r - cr^2\right) \left(2 - \frac{8}{3}r + 2cr^2 + 1\right)$$

$$= 4r - \frac{32}{9}r^2 - 3cr^2 + \frac{16}{3}cr^3 - 2c^2 r^4$$

yields the result. $\square$

**Corollary 17** (Lower Bound for Quadratic Minimization on Ellipsoids). [↓] *For any $d \geq 2$ and any horizon $T \in \mathbb{N}$, there exists an ellipsoid $\mathcal{E} \subset \mathbb{R}^d$, a strongly convex quadratic objective $f : \mathbb{R}^d \to \mathbb{R}$, and a starting point $x_0 \in \mathcal{E}$ such that the iterates $\{x_t\}_{t=0}^T$ generated by the FW algorithm with exact line search satisfy:*

$$f(x_t) - f(x^*) \geq \Omega\left(\frac{1}{t^2}\right) \quad \text{for all } t = 1, \ldots, T. \tag{35}$$

*Proof.* (Corollary 17) Fix $d \geq 2$ and $T \in \mathbb{N}$. We prove this by constructing a specific instance where the objective curvature aligns with the constraint geometry. Let $\mathcal{E} = \{x \in \mathbb{R}^d \mid x^\top A x \leq 1\}$ be an ellipsoid defined by any $A \succ 0$. We choose the objective function such that $Q = \alpha A$ for some $\alpha > 0$:

$$f(x) = \frac{1}{2} x^\top (\alpha A) x + c^\top x. \tag{36}$$

We reduce this problem instance $(\mathcal{E}, f)$ to the Euclidean unit ball instance $(\mathcal{B}, g)$ characterized in Theorem 6. While the FW algorithm is known to be affine-invariant, we prove the validity of this reduction here for the sake of completeness.

Since $A \succ 0$, the matrix square root $A^{1/2}$ is well-defined and invertible. Consider the linear bijection $\Phi : \mathbb{R}^d \to \mathbb{R}^d$ defined by $u = \Phi(x) = A^{1/2}x$. The image of the constraint set $\mathcal{E}$ under $\Phi$ is the unit ball $\mathcal{B}$:

$$
\begin{aligned}
x \in \mathcal{E} &\iff x^\top A x \leq 1 \\
&\iff (A^{-1/2}u)^\top A (A^{-1/2}u) \leq 1 \\
&\iff \|u\|_2^2 \leq 1.
\end{aligned} \tag{37}
$$

Let $g(u) := f(\Phi^{-1}(u)) = f(A^{-1/2}u)$. Substituting $Q = \alpha A$:

$$
\begin{aligned}
g(u) &= \frac{1}{2}(A^{-1/2}u)^\top (\alpha A)(A^{-1/2}u) + c^\top (A^{-1/2}u) \\
&= \frac{\alpha}{2} u^\top u + (A^{-1/2}c)^\top u \\
&= \frac{\alpha}{2} \|u\|_2^2 + \langle A^{-1/2}c, u \rangle.
\end{aligned}
$$

Completing the square by defining the target $\tilde{p} := -\frac{1}{\alpha}A^{-1/2}c$, we obtain:

$$
g(u) = \frac{\alpha}{2}\|u - \tilde{p}\|_2^2 - \frac{\alpha}{2}\|\tilde{p}\|_2^2. \tag{38}
$$

Therefore minimizing $f(x)$ on $\mathcal{E}$ is equivalent to minimizing $\frac{\alpha}{2}\|u - \tilde{p}\|_2^2$ on $\mathcal{B}$.

We now demonstrate that the FW iterates commute with $\Phi$. Let $x_t$ be the iterate in $\mathcal{E}$ and $u_t = A^{1/2}x_t$ be the corresponding point in $\mathcal{B}$.

*(i) Gradient Relation:*
$$
\begin{aligned}
\nabla_x f(x_t) = Qx_t + c &= \alpha A x_t + c \\
&= A^{1/2}\big(\alpha u_t + A^{-1/2}c\big) \\
&= A^{1/2}\nabla_u g(u_t).
\end{aligned} \tag{39}
$$

*(ii) LMO Equivalence:* The direction $s_t^{(x)}$ is computed as:

$$
\begin{aligned}
s_t^{(x)} &\in \arg\min_{z \in \mathcal{E}}\langle \nabla_x f(x_t), z \rangle \\
&= \arg\min_{z \in \mathcal{E}}\langle A^{1/2}\nabla_u g(u_t), z \rangle \\
&= A^{-1/2}\left( \arg\min_{v \in \mathcal{B}}\langle \nabla_u g(u_t), v \rangle \right) \quad \text{(via } z = A^{-1/2}v) \\
&= A^{-1/2}s_t^{(u)}.
\end{aligned}
$$

*(iii) Step Size:* The step size $\gamma_t$ minimizes the line search objective. Since $f(x_t + \gamma(s_t^{(x)} - x_t)) = g(u_t + \gamma(s_t^{(u)} - u_t))$, the optimal $\gamma_t$ is identical in both domains.

The sequence $\{u_t\}$ generated implicitly is exactly the FW trajectory for minimizing $\frac{\alpha}{2}\|u - \tilde{p}\|_2^2$ on $\mathcal{B}$. By Theorem 6, there exists a choice of $\tilde{p}$ and an initialization $u_0^{(T)}$ such that $g(u_t) - g(u^*) \geq \Omega(1/t^2)$ for all $t = 1, \dots, T$. Choosing $c = -\alpha A^{1/2}\tilde{p}$ and $x_0^{(T)} = A^{-1/2}u_0^{(T)}$ yields the same lower bound for $f(x_t) - f(x^*)$ for all $t = 1, \dots, T$. $\qquad\square$

