# OpenReview forum: "Lower Bounds for Frank-Wolfe on Strongly Convex Sets"
_ICML.cc/2026/Conference — ICML 2026 regular_

### Official Review · Reviewer_t7fZ · 2026-03-11

**Soundness:** 3
**Presentation:** 3
**Significance:** 3
**Originality:** 3
**Overall Recommendation:** 5
**Confidence:** 4

**Summary:**

**Summary**

This paper proposes a constructive procedure to find a worst-case function for the Frank-Wolfe method under a given stepsize rule. The idea is to leverage a toy projection problem, in which the Frank-Wolfe update reduces to a 2D scalar recursion. An $\Omega(1/\sqrt{\varepsilon})$ lower bound is constructed for a smooth, strongly convex objective.

**Compliance With Llm Reviewing Policy:**

Affirmed.

**Final Justification:**

After rebuttal, I keep my evaluation of the paper.

**Key Questions For Authors:**

**Minor issues**

1. Line 164

   I think $\theta_t$ is the cosine value instead of angle.

2. Line 181

   "any meaningful lower bound must assume a stepsize rule" should only apply to this example.

**Questions**

1. The conclusions state that the worst-case trajectory is unstable. Could you theoretically characterize this (in)stability?

**Limitations:**

Yes.

**Strengths And Weaknesses:**

**Strengths**

The paper is well-written and easy to follow. I like the simplicity of the approach and think it provides new tool for worst-case algorithm analysis.

**Weaknesses**

While the approach is novel to me, there are a few weaknesses that are worth further discussion in the revision.

1. Unlike the span-based information-theoretic complexity argument, the current lower-bound approach uses the specific form of the step-size rule. Moreover, it relies on a special starting point. This lower-bounding argument is algorithm-dependent and may not generalize easily. After the ICML submission timeline, some recent work has filled this gap [1].
2. I think the experiment (**Figure 2, 3**) at the beginning of **Section 3** is somewhat misleading since it seems to suggest a better complexity than $1/\sqrt{\varepsilon}$, which is not what you finally show. I guess you might add a magnified version of **Figure 2** around the constructed starting point.

Overall, I find the approach in the paper interesting from an algorithm analysis perspective. I recommend acceptance.

**References**

[1] Grimmer, B., & Liu, N. (2026). Lower Bounds for Linear Minimization Oracle Methods Optimizing over Strongly Convex Sets. *arXiv preprint arXiv:2602.22608*.

---

> ### Author Rebuttal · Authors · 2026-03-31
>
> We thank the reviewer for the positive assessment and feedback to our paper, we will address the mentioned issues below:
>
> ### Weaknesses
>
> > 1. Unlike the span-based information-theoretic complexity argument, the current lower-bound approach uses the specific form of the step-size rule. Moreover, it relies on a special starting point. This lower-bounding argument is algorithm-dependent and may not generalize easily. After the ICML submission timeline, some recent work has filled this gap [1].
>
> Thank you for pointing out this concurrent work, which appeared after the ICML submission deadline. We have added Grimmer and Liu (2026) to the revised version and discuss the complementary nature of our results.
>
> Please note that the construction of Grimmer & Liu (2026) also relies on a specific starting point ($x_0=0$), so this aspect is shared by both approaches. We believe the two contributions are complementary. Specifically, while the instance by Grimmer & Liu (2026) is in the high-dimensional setting, i.e. where the time horizon $T$ is smaller than the problem dimension $d$, our lower bound holds for any $d \ge 2$. Moreover, our result covers strongly convex **and** smooth sets, while their hard instance is non-smooth and they can only extend the result to "modestly" smooth sets for a reduced algorithm class. So while our result is more specific from an algorithmic angle, the problem setting is more general.
>
> > 2. I think the experiment (Figure 2, 3) at the beginning of Section 3 is somewhat misleading since it seems to suggest a better complexity than $1/\sqrt{\epsilon}$, which is not what you finally show. I guess you might add a magnified version of Figure 2 around the constructed starting point.
>
> Thank you for this suggestion. We have recomputed the heatmap in Figure 2 with a smaller threshold ($10^{-10}$ instead of $10^{-4}$), which is closer to the final primal gap of the trajectories in the other plots and makes the slow-convergence region more visible.
>
> The faster rates observed in Figure 3 are intentional: in the absence of a matching lower bound, they motivate the question of whether the $\mathcal{O}(1/\sqrt{\varepsilon})$ upper bound can be improved. The subsequent jump analysis shows that it cannot, and leads to the explicit construction of hard initializations in Figure 6. Contrasting these with random initializations completes the picture and confirms worst-case tightness of the $1/\sqrt{\varepsilon}$ rate.
>
> ### Questions
> > The conclusions state that the worst-case trajectory is unstable. Could you theoretically characterize this (in)stability?
>
> Our evidence for instability is numerical: the hard initialization requires very high precision. For example, over a sequence of 1000 steps one needs on the order of 1000 bits of precision to resolve the intended slow trajectory. Even small perturbations to the starting point or the step size cause the iterates to leave the slow trajectory and converge at the faster rate. This suggests that a smoothed complexity analysis could yield improved rates. We consider this a natural and interesting direction for future work.

---

> > ### Author Rebuttal · Reviewer_t7fZ · 2026-04-01
> >
> > Thank you for the response. I don't have further questions.

---

### Official Review · Reviewer_sN8A · 2026-03-14

**Soundness:** 3
**Presentation:** 3
**Significance:** 3
**Originality:** 3
**Overall Recommendation:** 5
**Confidence:** 2

**Summary:**

In this paper the authors provide a lower bound of solutions of convex constrained optimization problems using the Frank-Wolfe method. The Frank-Wolfe method is preferable compared to other methods like projected gradient descent in many setting because it avoids the use of projection oracles. In particular, the authors show that $\Omega(\sqrt{1/\epsilon})$ steps are required to compute \epsilon-approximate solutions to a constrained convex optimization problem even if the feasible set and the objective function are strongly convex. This matches the best known upper bound and hence settles the complexity of the problem.

**Compliance With Llm Reviewing Policy:**

Affirmed.

**Key Questions For Authors:**

1. You mention in the "Future work" paragraph that there is still a gap between the upper bound for convex objectives and the provided lower bound. Have you tried your method of finding worst-case initializations to simply convex objectives? E.g., $f(x) = ||x - p||_1$ instead of $f(x) = ||x - p||_2$ that you consider? It would be nice to know what is the bottleneck there and why your method does not work. It would also help the reader understand the limitations of your method.

**Limitations:**

See comments and questions.

**Strengths And Weaknesses:**

Strengths
-----------------------

- This is a clean result and answers an interesting question.

- The paper is well-written.

- The proof technique in novel to the best of my knowledge and relies on finding initializations that are useful both theoretically to prove lower bounds and in practice for benchmarking algorithms.


Weaknesses - Comments
-----------------------------------
1. I guess the second footnote in page 3 should be x^* instead of x? Otherwise, I am very confused with what this table says.

2. The main idea of the proof relies on constructing a worst-case initialization. It is hence unclear if a random initialization could help bypass the $1/\sqrt{\epsilon}$ lower bound. Of course, the Garber & Hazan (2015) upper bound also applies for worst-case initializations, so Theorem 6 shows the tightness of this method. Nevertheless, it still remains an interesting open problem if random initializations could help for all instances. In the examples considered in this paper random initialization actually helps as illustrated in Figure 6.

---

> ### Author Rebuttal · Authors · 2026-03-31
>
> We thank the reviewer for the careful reading, positive assessment, and constructive comments.
>
> ### Weaknesses - Comments
>
> > 1. I guess the second footnote in page 3 should be x^* instead of x? Otherwise, I am very confused with what this table says.
>
> Yes, indeed you are right that it should be $x^*$, we have fixed this in the revised version.
>
> > 2. The main idea of the proof relies on constructing a worst-case initialization. It is hence unclear if a random initialization could help bypass the $1 / \sqrt{\epsilon}$ lower bound. Of course, the Garber & Hazan (2015) upper bound also applies for worst-case initializations, so Theorem 6 shows the tightness of this method. Nevertheless, it still remains an interesting open problem if random initializations could help for all instances. In the examples considered in this paper random initialization actually helps as illustrated in Figure 6.
>
> For our instance, the hard starting points are very specific: for example, starting points $x_0$ that are collinear with the optimal point $p$ cause the method to converge in a single step, and more generally, even small perturbations to the constructed initialization cause the iterates to leave the slow trajectory and converge at the faster rate. The hard initialization requires roughly $T$ bits of precision to sustain the slow convergence over $T$ steps, which makes it unlikely that a random initialization would land on it. This instability suggests that a smoothed complexity analysis could yield improved rates for our instance, and we consider this an interesting direction for future work.
>
> ### Key Questions for Authors:
> > You mention in the "Future work" paragraph that there is still a gap between the upper bound for convex objectives and the provided lower bound. Have you tried your method of finding worst-case initializations to simply convex objectives? E.g., $f(x) = \\|x-p\\|_1$ instead of $f(x) = \\|x-p\\|_2$ that you consider? It would be nice to know what is the bottleneck there and why your method does not work. It would also help the reader understand the limitations of your method.
>
> Yes, we explored a direct translation to the merely convex setting by replacing
> $f(x)=\\|x-p\\|_2^2$ with $\tilde f(x)=\\|x-p\\|_2$. In our construction, this change does not alter the FW direction (the LMO is invariant to positive rescaling of the gradient), and exact line search selects the same minimizers along each FW segment. Therefore, we obtain exactly the same $(r,s)$ dynamics as in the original setting. However, the suboptimality becomes linear in the radius variable (instead of quadratic), which yields an $\Omega(1/\epsilon)$ lower bound from this argument.
>
> The limitation is that $\tilde f$ is merely convex and not smooth at $x=p$ (since it is not differentiable). The standard FW rate in the non-smooth setting is $\mathcal{O}(1/\epsilon^2)$, so an $\Omega(1/\epsilon)$ argument from this reduction cannot be tight.
>
> The suggested $\\|x-p\\|_1$ is potentially an interesting choice, as the entries of the gradient satisfy $(\nabla f(x))_i = \pm 1$ if $(x-p)_i \neq 0$, resulting in only a selected number of extreme points. The iterates no longer stay in a two-dimensional plane, and one would have to derive a completely new dynamic. Nonetheless, given the finite number of extreme points, this might be worth analyzing in future work, as it could shed further light on whether more robust worst-case initializations exist in the merely convex setting.

---

> > ### Author Rebuttal · Reviewer_sN8A · 2026-04-03
> >
> > Thank you for the answers. My questions are resolved.

---

### Official Review · Reviewer_6M8u · 2026-03-15

**Soundness:** 3
**Presentation:** 3
**Significance:** 2
**Originality:** 3
**Overall Recommendation:** 4
**Confidence:** 3

**Summary:**

This paper studies whether the known (O(1/\sqrt{\varepsilon})) convergence guarantee for vanilla Frank–Wolfe on smooth and strongly convex problems is tight. It analyzes a simple quadratic-over-ball model, identifies hard trajectories numerically, and then derives a matching lower bound. The main takeaway is that even under smoothness and strong convexity assumptions, vanilla FW may still require (\Omega(1/\sqrt{\varepsilon})) iterations.

**Compliance With Llm Reviewing Policy:**

Affirmed.

**Final Justification:**

Thank you for the author's feedback; I will maintain my score.

**Key Questions For Authors:**

Please refer to what is described in "Strengths And Weaknesses".

**Limitations:**

The main weakness is the empirical observation section, followed by the limited experimental support. Overall, this is a theory-strong but experiment-weak paper.

**Strengths And Weaknesses:**

Soundness
The theoretical part appears solid, and the main result is reasonably convincing. The main weakness is not the theory, but the empirical observation and experimental section: these parts are more heuristic and less fully validated, so the evidence there feels weaker than the theory.

Presentation
The paper is generally clear and well organized. The overall narrative is easy to follow, and the related-work positioning is mostly adequate. That said, the boundary between numerical observations and formal theoretical conclusions could be stated more carefully.

Significance
The paper addresses a meaningful theoretical question about a classical optimization method. Its contribution is more about improving our understanding of vanilla Frank–Wolfe than introducing a new practical method, but that is still valuable and relevant for the field.

Originality
The work has clear originality, especially in how it combines numerical observations, dynamical analysis, and an analytic lower-bound argument. It does not propose a new algorithm, but it provides a new perspective and a sharper understanding of an existing theoretical question.

---

> ### Author Rebuttal · Authors · 2026-03-31
>
> We thank the reviewer for the careful reading and constructive feedback.
>
> ### Strengths and Weaknesses
>
> > Soundness The theoretical part appears solid, and the main result is reasonably convincing. The main weakness is not the theory, but the empirical observation and experimental section: these parts are more heuristic and less fully validated, so the evidence there feels weaker than the theory.
>
> We understand the concern that the empirical part of the submission appeared more heuristic than the theoretical section. Our numerical experiments are fully deterministic (there is no randomness in the construction or optimization process), so here “validation” refers not to statistical significance, but to the consistency of the observed behavior across instances and its alignment with the theoretical picture. To make this clearer, we added a new figure that compactly summarizes the empirical phenomena used in our discussion. It shows the hard-instance trajectory together with the threshold curve $g(r)$ and the affine lower bound $s = 1 - \tfrac{4}{3}r$ in the $(r,s)$ plane. This plot visualizes the central geometric claims of Section 3: the trajectory stays below $g(r)$, exhibits monotone contraction, and remains above the affine approximation. We also revised the text to state explicitly that these experiments serve as deterministic, illustrative evidence that motivates our analytic lower bound, which is itself established purely theoretically.
>
> > Presentation The paper is generally clear and well organized. The overall narrative is easy to follow, and the related-work positioning is mostly adequate. That said, the boundary between numerical observations and formal theoretical conclusions could be stated more carefully.
>
> We appreciate the concern regarding the boundary between the numerical observations and the theoretical lower-bound proof. While these parts are already presented in separate sections, we make the distinction even more explicit in the revised version. In particular, the new plot introduced above clearly illustrates the numerical observations discussed in Section 3. Additionally, we have revised the jump analysis by introducing a formal definition of jumps and a necessary condition for their occurrence, which provides a concrete theoretical rationale for constructing monotone sequences and makes the connection to the proof strategy more transparent.
>
> > Significance The paper addresses a meaningful theoretical question about a classical optimization method. Its contribution is more about improving our understanding of vanilla Frank–Wolfe than introducing a new practical method, but that is still valuable and relevant for the field.
>
> We thank the reviewer for this assessment, which matches our intended scope. Our goal is to characterize the optimal convergence rate of vanilla FW (with standard short-step / exact line-search step sizes), rather than to introduce a new algorithmic variant.
>
> > Originality The work has clear originality, especially in how it combines numerical observations, dynamical analysis, and an analytic lower-bound argument. It does not propose a new algorithm, but it provides a new perspective and a sharper understanding of an existing theoretical question.
>
> We agree with this characterization and will keep the novelty statement concise and aligned with this scope in the final version.

---

> > ### Author Rebuttal · Reviewer_6M8u · 2026-04-01
> >
> > Thank you for the clear and thoughtful rebuttal. The response satisfactorily addresses my main concerns, especially regarding the theoretical part and the intended role of the numerical observations in the paper. However, I still do not feel that my concerns about the empirical section have been fully resolved, in particular with respect to how strongly the experiments support the overall claims. Since my original assessment was already positive and remains unchanged, I do not plan to update my score. That said, I may lower my confidence slightly to reflect the remaining uncertainty on the empirical side.

---

> > > ### Author Response · Authors · 2026-04-04
> > >
> > > Thank you for clarifying your remaining concern regarding the empirical section. To further contextualize our theoretical results, we conducted additional experiments. The results are available at the following anonymized link:
> > > https://osf.io/9hsg8/overview?view_only=6a574830d08a48e19f0f3a4b59c52379  (two plots: fw_convergence_all_kappa.pdf  and fw_convergence_all_d.pdf)
> > >
> > > We ran Frank-Wolfe with the short-step on strongly convex quadratic objectives over the unit Euclidean ball, varying both the dimension $d$ and the condition number $\kappa$ (max eigenvalue/ min eigenvalue of the Hessian). We consider three regimes, depending on the location of the unconstrained minimizer: in the interior of the ball, on the boundary, and outside the ball. In each case, we plot the primal gap versus the iteration number on a log-log scale. The thick curves correspond to initialization at the origin, while the thin transparent curves show additional runs from random feasible starting points.
> > >
> > > Across these settings, we consistently observe an initial convergence regime that closely follows a $t^2$-type decay. However, depending on the problem parameters, this behavior eventually transitions to a faster regime later in the optimization process. In particular, larger condition numbers prolong the initial slow phase before this transition occurs.
> > >
> > > This empirical pattern was one of the motivations for our work: while the early-stage behavior might suggest a $1/t^2$-type rate, it is not a priori clear whether this reflects an intrinsic worst-case limitation or merely a transient phenomenon. Our theoretical result resolves this question by showing that such slow convergence can indeed persist in the worst case.

---

### Official Review · Reviewer_SNLR · 2026-03-22

**Soundness:** 2
**Presentation:** 2
**Significance:** 2
**Originality:** 3
**Overall Recommendation:** 4
**Confidence:** 4

**Summary:**

The paper establishes a new $\Omega(\frac{1}{\sqrt{\epsilon}})$ lower bound for the **classical Frank-Wolfe (FW) method with exact line search** for minimizing a simple strongly convex quadratic function over a standard Euclidean ball:
$$
\min_{x \in B_1(0)} \\{ f(x) = \\| x - p \\|^2 \\},
$$
where $B_1(0) = \\{ x \in \mathbb{R}^d \colon \\| x \\| \leq 1 \\}$, $p \in \mathbb{R}^d$ is a given unit vector, and $\epsilon > 0$ is the desired accuracy in terms of $f(x)$.

An immediate consequence of this result is that the same lower bound holds for the same algorithm but applied to a more general family of $\mu$-strongly convex and $L$-smooth functions over an $\alpha$-strongly convex and $\beta$-smooth compact feasible set with $\mu = L = 2$ and $\alpha = \beta = 1$.

**Compliance With Llm Reviewing Policy:**

Affirmed.

**Final Justification:**

My initial concern about the usefulness of the obtained results has been completely resolved. Therefore, I have increased my original score by one point.

The other big concern about unclear mathematical presentation and missing proof details still remains valid. Fixing the issues outlined in my review requires a major revision followed by an additional round of review, which the conference format does not allow for.

**Key Questions For Authors:**

1. The definition of a strongly convex set in lines 119-121 appears rather nonstandard. Is it equivalent to a more classical definition stating that, for any two points $x, y$ in the set and any point $z \in [x, y]$, the set also contains a ball centered at $z$?

2. What is the precise mathematical meaning of "corresponds" in Lemma 8?

3. Does the FW method with exact line search studied in this paper guarantee the $O(\frac{1}{\sqrt{\epsilon}})$ complexity for a general problem of minimizing a strongly convex and smooth function over a strongly convex and smooth set, as the FW method from Garber & Hazan (2015)?

4. It seems that the "jump analysis" in Section 3 is completely redundant and is not used at all in the proof of Theorem 6. Is it correct? If so, why presenting it?

**Limitations:**

Yes.

**Strengths And Weaknesses:**

## General comments

The paper is interesting, and the established lower bound appears to be new (at least, to my knowledge).

However, the **usefulness of the obtained lower bound raises some concerns**. If we knew that the algorithm under consideration does attain a $O(\frac{1}{\sqrt{\epsilon}})$ convergence rate for minimizing a general strongly convex and smooth function over a strongly convex and smooth feasible set, then the obtained lower bound could be used for confirming the optimality of the already known upper bound. However, to my knowledge, there is no such an upper bound for the version of FW studied in this paper. Instead, the $O(\frac{1}{\sqrt{\epsilon}})$ is achieved by some other versions of FW, e.g., the one from Garber & Hazan (2015), and the current paper does not establish any lower bounds for those other versions.

Additionally, in terms of presentation, the **paper appears to be confusing in several key places** and **some proofs miss important details**. This makes it difficult to check the correctness of the constructed lower bound raising additional concerns about its validity. For more details, please see the list of "major remarks" below.

## Major remarks

1. **Unclear main result.** The statement of the main result in Theorem 6 is unclear. In particular, it is unclear what exactly $\Omega(\frac{1}{t^2})$ means and which constant it hides inside. After a closer inspection of the proof, one can see that the precise statement is actually the squared version of (15):
   $$
   f(x_t) - f^* \geq \frac{r_0^2}{(1 + \frac{8}{3} r_0 t)^2}, \quad t = 1, \ldots, T,
   $$
   where $r_0^2 = f(x_0) - f^*$ is the initial function suboptimality. However, this result is rather unsatisfying because it does not say anything about how small $r_0$ could be. Instead, the current statement simply claims that, for each $T$, there exists a feasible point $x_0$, and does not rule out the possibility of $r_0$ being a very small function of $T$. If this is the case, then the claimed $\Omega(\frac{1}{t^2})$ complexity is actually misleading.

2. **Unclear meaning of "parametrization".** To arrive at the main result, the paper introduces a couple of different "coordinate representations" or "parametrizations" of the FW iterates $x_t$. It starts with the "polar coordinates" $(r_t, \theta_t)$ in Section 2 and then switches to the $(r_t, s_t)$ "parametrization" in Section 3. The proof of the main result crucially relies on the fact that "the pair $(r_t, \theta_t)$ *fully characterizes* $x_t$" (line 165) and that "any point $(r, s) \in M$ *corresponds* to a feasible point $x \in B_1(0)$" (Lemma 8). However, it is unclear what precisely "fully characterizes" or "corresponds" mean mathematically. Does it mean that there is a one-to-one correspondence between all feasible $(r, \theta)$ (or $(r, s)$) and all points $x \in B_1(0)$, or does it mean something else? What is the corresponding transformation mapping $(r, \theta)$ (or $(r, s)$) into $x$ and vice versa? The paper does not discuss any of these important questions.

3. **Unclear logic.** The logic in several places is unclear and difficult to follow. In particular:

   1. Line 221 claims that "... since $r_t$ goes to zero, ... the contraction factor needs to be strictly decreasing, i.e., $s_{t + 1} < s_t$, to allow for a jump", but it is unclear how exactly (8) (which is an inequality rather than an identity) rules out the possibility of $s_{t + 1} \geq s_t$ for a certain $t$. Additionally, it is not fully clear what a "jump" actually is (how it is defined).

   2. Line 245 states that "The result in Proposition 5 shows that jumps only occur in the unstable regime". However, it is unclear how exactly Proposition 5 implies that "jumps occur in the unstable regime" (and what it actually means).

   3. Proof of Lemma 11: The argument in lines 1015-1056 is unclear. For example, what does (7) have to do with the claim in Lemma 11 given that there is no index $t$ in the statement of the lemma at all? Instead, the statement simply states that $F(G(r, s)) = (r, s)$, so why not directly check that? The second part of the proof also overloads the $(r, s)$ and $(r', s')$ notation from the first part, which is quite confusing.

4. **Missing numerical results.** In several places, the paper discusses certain important numerical results as if they have just been presented, while, in reality, they are nowhere to be found. It seems that the authors forgot to include them in the paper. For example:

   1. Lines 321-327 discuss that "the trajectories $(r_t, s_t)$ stay closely below the threshold $g(r)$", but no corresponding plot is provided. Further, it is stated that "linear approximation yields $s \approx 1 - \frac{4}{3} r$ for small $r$", but again no justification (or numerical evidence) is provided.

   2. Line 275 says that "$s \geq 1 - \frac{4}{3} r$ is empirically motivated", but no such empirical results are presented. A similar comment applies to lines 289-291, 377 and 385-387.

   3. Lines 306-309 state that "we have already observed this non-injectivity of the forward dynamics in the grid search", but no corresponding results have been shown earlier.

   4. Lines 906 and 922 say that "The maximum length of these peaks is determined by the resolution of the grid search" and "In high-precision runs, it produces monotone segments that stay on the stable trajectory for arbitrarily long times", but there are no plots confirming these claims.

1. **Missing details in proofs.** Several proofs miss important details and simply state something without any justification. For example:

   1. Proof of Lemma 13: Why is $1 - \frac{4}{3} r + c r^2 \leq \frac{1}{1 + r}$ (line 1088)? Why do the inequalities for $X$ and $Y$ in lines 1098 and 1110 actually hold?

   2. Proof of Lemma 14: It is simply written that the inequalities in lines 1138 and 1152 "follow from bounding the higher order terms", but no details are provided. Additionally, there is no justification of the identities in lines 1133 and 1147, involving polynomials of degree 9 and complicated coefficients such as $\frac{5759}{729}$; instead, they are simply presented as "obvious calculations". Similarly, no proofs are given for the final inequalities in lines 1164 and 1174.

   3. Proof of Theorem 6: Line 1241 applies the inequality $\frac{1}{1 - x} \leq 1 + 2 x$ valid for $x \in (0, \frac{1}{2}]$ substituting $x = \frac{4}{3} u_{t + 1}$. However, no justification is provided as to why this $x$ satisfies $x \leq \frac{1}{2}$.

   4. Definition 7 simply states that $F(r, s)$ defined by (23) takes values in $M$ whenever $(r, s) \in M$ but no proof is given.

   5. First paragraph in Section B.2: It is simply written that $s_{t + 1} \geq s_t$ is equivalent to line 823 but no justification is provided.

2. **Redundant results.** A few results in the paper do not seem to be used anywhere and therefore appear redundant, e.g., Proposition 5, Corollary 15, and the "second branch" in the domain $M$ from Definition 7 and eq. (34). It is therefore unclear what their value is and why they are presented at all.

## Other remarks

1. There is a mistake in line 815: one cannot estimate $\sqrt{(s^2 - 1) (s^2 (1 + r)^2 - 1)} \leq 0$ since otherwise it necessarily implies that $s = \frac{1}{1 + r}$, while one only knows that $0 \leq s \leq \frac{1}{1 + r} < 1$.

2. Paragraph after Lemma 1: 1) One needs to require that $x_0 \neq 0$ so that $\mathrm{span}\\{ p, x_0 \\}$ is indeed a plane. 2) It would be helpful to clarify that the origin of "polar coordinates" is placed at $p$ and provide a picture illustrating the construction. 3) It would be more convenient to define $\theta_t = \frac{\langle p - x_t, p \rangle}{\\| p - x_t \\|}$ so that $\theta_t$ is nonnegative (rather than nonpositive). 4) One needs to require that $x_t \neq p$.

3. Line 613, "if $x_t = p$, then $v_t = 0$". Why is that? It should then be in the definition of the algorithm?

4. Lines 793-800: Monotonicity of these functions is obvious, no need to compute any derivatives.

5. Lines 858-870: This can be done much simpler by estimating $s_t \leq \frac{1}{1 + r}$ in the original expression.

6. Definition 9 and later: The $X$ and $Y$ notation is rather confusing. First, it should at least be $X(r, s)$ and $Y(r, s)$. Second, everything depends on the sum $s' \equiv X + Y$ so why not introduce directly $s'(r, s)$?

7. Lines 945-947: It would be helpful to elaborate on this.

8. Lines 1003-1013: In was already shown in the proof of Lemma 10 that $s' = X + Y$ is decreasing in $s$. Therefore, the inequality $X + Y + r \leq 1$ immediately follows by substituting $s = 0$.

## Typos

1. Eq. (7): $s_{t + 1}^2 = \frac{r_{t + 2}^2}{r_{t + 1}^2}$.

2. Line 778: $\sqrt{\frac{2 - r}{4}}$.

3. Line 850: Remove "$\subseteq [0, \frac{1}{1 + r_t}]$".

4. Line 918: if $\tau(m)$ and $\tau(u)$ have the same parity.

5. Lemma 10: $\geq \frac{5}{12}$.

6. Line 1250: $\tilde{M}_{t + 1}$.

---

> ### Author Rebuttal · Authors · 2026-03-31
>
> We thank the reviewer for the detailed feedback, especially on the proof section. Below we address the main points.
> ### General comments
> Garber & Hazan (2015) analyze vanilla FW with the *short-step rule*, which is covered by our results. They define
> $$
> \gamma_t = \arg\min_{\gamma \in [0,1]}\gamma \langle v_t-x_t, \nabla f(x_t) \rangle+\gamma^2\frac{L}{2} \\|v_t-x_t\\|^2.
> $$
> Here we adapted their notation to our paper, using $L$ for the smoothness constant and $v_t$ for the LMO output.
> The above minimization problem has a closed-form solution,
> $$
> \gamma_t = \min \left\\{ \frac{\langle x_t-v_t, \nabla f(x_t) \rangle}{L\\|v_t-x_t\\|^2}, 1 \right\\},
> $$
> which is exactly the one we state in our paper. Their proof uses a per-iteration progress bound that also holds for exact line search, so their \(\mathcal{O}(1/\sqrt{\varepsilon})\) upper bound applies to the exact vanilla FW variant we study. Our \(\Omega(1/\sqrt{\varepsilon})\) lower bound matches this rate. Thus the \(\varepsilon\)-dependence for vanilla FW with short-step or exact line search on smooth, strongly convex problems is optimal. We revised the introduction to clarify this.
> ### Major remarks
> **Unclear main result.**
> This is an important point. In the submitted version, the theorem statement did not explicitly rule out a dependence of the initial residual $r_0$ on the horizon $T$. We revised the proof and use a residual threshold $r_{\max} = 1/10$ as the termination criterion in the backward reconstruction algorithm. As a consequence, we perform $\hat T \geq T$ backward steps to retrieve the starting point. With this modification we prove the uniform bound $1/18\le r_0 \le 1/10$ which is independent of $T$. Thus the $\Omega(1/t^2)$ lower bound holds with a horizon-independent constant and is not driven by a vanishing initialization.
>
> **Unclear meaning of "parametrization".**
> There is no global one-to-one correspondence. The iterate sequence $\\{x_t\\}_t$, and hence $\\{r_t\\}_t$ and $\\{s_t\\}_t$, is uniquely determined by the FW update rule, the step-size rule, and the starting point $x_0$. Thus each $x_t$ induces exactly one pair $(r_t,s_t)$. The converse is not unique: the angle in polar coordinates is defined only up to symmetry, and the quadratic equation in Lemma 8 that recovers $\theta$ from a given $s$ has for $s \ge \frac{1}{2}$ two solutions. We revised the corresponding paragraphs and now state this relation more formally. This does not affect the main result, it only shows that multiple branches can generate hard instances.
>
> **Unclear logic (jump discussion).**
> We revised this section to make the logic explicit. First, jumps are now formally defined as iterations with $s_t < \frac12$. Second, we replaced the previous sufficient jump characterization with a necessary condition. The new result directly connects the occurrence of jumps to monotonicity of the contraction factors. This clarifies why we construct sequences with monotone increasing contraction rates and makes the motivation more theoretically grounded.
>
> **Missing numerical results.**
> The reviewer is right that these statements need direct supporting evidence. We therefore added a second version of Figure 4 that includes the $(r,s)$ trajectory of a hard instance together with the affine approximation $s = 1-\frac43 r$. This makes the key geometric claims visible in a single plot: the $(r,s)$ trajectory of the hard instance has monotone increasing contraction rates and remains below the monotonicity threshold $g(r)$ while staying above the affine approximation $s = 1-\frac43 r$.
> ### Missing details in proofs, other remarks and typos
> We thank the reviewer for their thorough proof comments and use them to systematically polish the technical parts: we added missing intermediate steps, clarified overloaded notation, and corrected the noted typos; all changes are reflected in the revised manuscript.
> ### Key questions
> 1. Yes, these definitions are equivalent (see Vladimir V. Goncharov and Grigorii E. Ivanov, *Strong and Weak Convexity of Closed Sets in a Hilbert Space*, Theorem 2.1).
> 2. We revised Lemma 8 and now state this more clearly: for each $(r,s) \in M$, there exists an $x\in B_1(0)$ with $r=\\|x-p\\|$ and $s=\frac{\\|x'-p\\|}{\\|x-p\\|}$, where $x'$ is the succeeding FW iterate.
> 3. Yes. See our response to "General comments" for more details.
> 4. The jump analysis is not used in the formal proof of Theorem 6, but it plays a key motivational role: it shows that any trajectory with non-monotone contraction rates must eventually "jump" and recover a faster rate, which is precisely why hard instances require monotone contraction sequences. This insight guided the construction of the hard initialization. In the revision, we replaced the sufficient jump condition with a necessary condition that directly proves the contraction rate must strictly decrease before a jump. We also made the distinction between the motivational Section 3 and the formal proof in Section 4 explicit.

---

> > ### Author Rebuttal · Reviewer_SNLR · 2026-04-03
> >
> > Dear authors, thank you for your reply.
> >
> > **Regarding Garber & Hazan (2015).**
> > Thank you for the clarification. I somehow missed it, but indeed, their algorithm is exactly what is referred to as the short-step FW method in your paper (please do clarify this in the revised version, as well as your other comment on the version with exact line search). This means that **my original concern about the usefulness of the obtained results has been fully resolved. I therefore increase my original score by one point.**
> >
> > **Regarding confusing presentation and missing proof details.**
> > The issues with the mathematical presentation I outlined in my original review are quite substantial. Addressing them properly requires a major revision followed by an additional round of review. Since the conference submission format does not allow any additional rounds of review, **I do not consider these concerns to be resolved** based solely on your commitment to do so.
> >
> > **Regarding the definition of a strongly convex set.**
> > I have checked Theorem 2.1 from (Vladimir V. Goncharov and Grigorii E. Ivanov, Strong and Weak Convexity of Closed Sets in a Hilbert Space). However, among many equivalent definitions (a)-(s) listed there, I did not find the one from your paper. When preparing a revision, please either replace your definition with a more classical one, or explain in detail the relationship between your definition and a more classical one.

---

> > > ### Author Response · Authors · 2026-04-05
> > >
> > > Thank you for the increased score. We edited this reply because we intended to post a detailed follow-up on the remaining presentation concerns, but did not realize we could not add a second reply. Below we detail the critical fixes in the revised manuscript.
> > >
> > > **1. Unclear main result (Major remark 1): the $r_0$-dependence on $T$ is eliminated.**
> > >
> > > With a simple argument one can remove the potential $T$-dependency of $r_0$. The original proof ran exactly $T$ backward steps, coupling $r_0$ to $T$. The revised proof uses a *threshold-based* backward reconstruction with $r_{\max} = \frac{1}{10}$. Starting from the terminal point $(\varepsilon,\, 1 - \frac{4}{3}\varepsilon + 2\varepsilon^2)$ with $\varepsilon = \frac{1}{10 + \frac{8}{3}T}$, we apply the backward dynamics $G$ until the residual first exceeds $r_{\max}$. The stopping index $\widehat{T}$ satisfies $\widehat{T} \geq T$ (shown by telescoping $\frac{1}{u_{t+1}} \geq \frac{1}{u_t} - \frac{8}{3}$, which gives $1/u_T \geq 1/u_0 - \frac{8}{3}T$ and since $u_0=\varepsilon$ we get $u_T \leq \frac{1}{10}$).
> > >
> > > The starting point is $(r_0, s_0) = (u_{\widehat{T}},\, v_{\widehat{T}})$, the *last* iterate below the threshold. We then bound $r_0$ away from $0$ as follows. Lemma 14 shows that each backward step preserves the invariant $v_t = 1 - \frac{4}{3}u_t + c_t u_t^2$ with $c_t \in [1, \frac{5}{2}]$. Applying this at index $\widehat{T}+1$ and using $c_{\widehat{T}+1} \geq 1$ gives
> > >
> > > $$v_{\widehat{T}+1} \geq 1 - \tfrac{4}{3}u_{\widehat{T}+1} + u_{\widehat{T}+1}^2 = \left(u_{\widehat{T}+1} - \tfrac{2}{3}\right)^2 + \tfrac{5}{9} \geq \tfrac{5}{9}.$$
> > >
> > > Since $u_{\widehat{T}+1} \geq \frac{1}{10}$ by definition of $\widehat{T}$ and $r_0 = u_{\widehat{T}} = v_{\widehat{T}+1} \cdot u_{\widehat{T}+1}$, we get $r_0 \geq \frac{5}{9} \cdot \frac{1}{10} = \frac{1}{18}$. The final bound
> > >
> > > $$f(x_t) - f(x^*) \geq \frac{r_0^2}{\left(1 + \frac{8}{3} r_0 t\right)^2}$$
> > >
> > > then holds with $\frac{1}{18} \leq r_0 \leq \frac{1}{10}$ uniformly in $T$, yielding a genuine $\Omega(1/t^2)$ lower bound. This argument is fully spelled out in the revised proof of Theorem 6 (Appendix C).
> > >
> > > **2. Parametrization (Major remark 2): why non-injectivity does not affect the result.**
> > >
> > > There is no global bijection between $(r,s) \in M$ and $x \in B_1(0)$, and the proof does not require one. It only uses the *forward direction*: the revised Lemma 8 now states explicitly that for any $(r,s) \in M$, there exists $x \in B_1(0)$ with $\|x-p\| = r$ and $\frac{\|x'-p\|}{\|x-p\|} = s$, where $x'$ is the FW iterate from $x$. The proof reconstructs $\theta$ from $(r,s)$ by solving a quadratic and verifying feasibility $\theta \leq -\frac{r}{2}$.
> > >
> > > The backward reconstruction builds a sequence $\{(u_t, v_t)\} \subset \widetilde{M} \subset M$, and this lemma converts each pair into a feasible $x_t$. Since the proof only needs *existence* of one feasible realization per $(r,s)$ pair, non-uniqueness of the map $(r,s) \mapsto x$ does not pose a problem. Concretely, the quadratic for $\theta$ has two roots, and we show that the negative root $\theta = -s^2(r+1) - \sqrt{(s^2-1)(s^2(1+r)^2-1)}$ always satisfies the feasibility constraint $\theta \leq -\frac{r}{2}$, so one knows a priori which root to select. Additionally, we added a lemma proving $F\colon M \to M$, previously stated without proof. The key step is $(1 - (1+r)s(1+rs))^2 \geq 0$, implying $s_+ \leq \frac{1}{1+r_+}$. In the other direction, $F$ is not injective: as now discussed in the revised proof of Lemma 11, there are two preimage branches and $G$ consistently selects one. This does not affect the lower bound.
> > >
> > > **3. Key proof details now filled in (Major remark 5).**
> > >
> > > - *Lemma 13 (feasibility):* $1 - \frac{4}{3}r + cr^2 \leq \frac{1}{1+r}$ for $c \in [1,\frac{5}{2}]$, $r \in (0,\frac{1}{10}]$ now proved via $s < 1 - \frac{4}{3}r + \frac{r}{4} < 1 - r \leq \frac{1}{1+r}$, using $cr \leq \frac{1}{4}$.
> > > - *Theorem 6, $\frac{1}{1-x} \leq 1+2x$ bound:* Now justified explicitly: $x = \frac{4}{3}u_{t+1} \leq \frac{4}{3} \cdot \frac{1}{10} = \frac{2}{15} < \frac{1}{2}$.
> > > - *Lemma 8 ($\theta \geq -1$):* We write $\theta = -\langle (s,\, \sqrt{1-s^2}),\, (s(r+1),\, \sqrt{1-s^2(1+r)^2}) \rangle$ and apply Cauchy–Schwarz, giving $\theta \geq -\|(s, \sqrt{1-s^2})\| \cdot \|(s(r+1), \sqrt{1-s^2(1+r)^2})\| = -1$.
> > >
> > > **4. Strongly convex set definition.**
> > > This was an error on our part. We will replace it with the classical definition (intersection of balls). This does not affect the result, as the definition was not used in the proof.
> > >
> > > **5. Jump analysis (Major remark 3).**
> > >
> > > Jumps are now formally defined as iterations with $s_t < \frac{1}{2}$. We replaced the old sufficient condition with a necessary one: if $s_{t+1} < \frac{1}{2}$ and $r_t < \sqrt{2} - 1$, then $s_t > s_{t+1}$. This directly shows contraction rates must decrease before any jump, motivating the monotone construction.

---

### Decision · Program_Chairs · 2026-04-30

**Decision:**

Accept (regular)

**Comment:**

This paper establishes a constructive lower bound for the Frank-Wolfe method under smooth and strongly convex objectives and constraint sets. The contributions are primarily of theoretical interest. The analysis technique is quite original. The reviews were largely positive and I also agree with this overall assessment. Reviewer SNLR has provided valuable suggestions to improve the presentation and clarity, which should be addressed in the final version.